# Graph Lottery Ticket Automated

**Guibin Zhang**[1,2,†], **Kun Wang**[3,†], **Wei Huang**[4], **Yanwei Yue**[1], **Yang Wang**[3],
**Roger Zimmermann**[5], **Aojun Zhou**[6], **Dawei Cheng**[1], **Jin Zeng**[1,*], **Yuxuan Liang**[1*]

[1]Tongji University    [2]The Hong Kong University of Science and Technology (Guangzhou)
[3]University of Science and Technology of China (USTC)    [4]RIKEN AIP
[5]National University of Singapore    [6]The Chinese University of Hong Kong

## Abstract

Graph Neural Networks (GNNs) demonstrate superior performance in various graph learning tasks, yet their wider real-world application is hindered by the computational overhead when applied to large-scale graphs. To address this issue, the Graph Lottery Ticket (GLT) hypothesis assumes that GNN with random initialization harbors a pair of core subgraph and sparse subnetwork, which can yield comparable performance and higher efficiency to that of the original dense network and complete graph. Despite that GLT offers a new paradigm for GNN training and inference, existing GLT algorithms heavily rely on trial-and-error pruning rate tuning and scheduling, and adhere to an irreversible pruning paradigm that lacks elasticity. Worse still, current methods suffer scalability issues when applied to deep GNNs, as they maintain the same topology structure across all layers. These challenges hinder the integration of GLT into deeper and larger-scale GNN contexts. To bridge this critical gap, this paper introduces an **A**daptive, **D**ynamic, and **A**utomated framework for identifying **G**raph **L**ottery **T**ickets (`AdaGLT`). Our proposed method derives its key advantages and addresses the above limitations through the following three aspects: 1) tailoring layer-adaptive sparse structures for various datasets and GNNs, thus enabling it to facilitate deeper GNNs; 2) integrating the pruning and training processes, thereby achieving a dynamic workflow encompassing both pruning and restoration; 3) automatically capturing graph lottery tickets across diverse sparsity levels, obviating the necessity for extensive pruning parameter tuning. More importantly, we rigorously provide theoretical proofs to guarantee `AdaGLT` to mitigate over-smoothing and obtain improved sparse structures in deep GNN scenarios. Extensive experiments demonstrate that `AdaGLT` outperforms state-of-the-art competitors across multiple datasets of various scales and types, particularly in scenarios involving deep GNNs.

## 1 Introduction

Graph Neural Networks (GNNs) have emerged as the prevailing model for graph representation learning tasks (Kipf & Welling, 2017b; Veličković et al., 2018; Hamilton et al., 2017; Zhang & Chen, 2019; Liang et al., 2023; Gao et al., 2023; Cheng et al., 2021; Duan et al., 2024; Zhang et al., 2024). The success of GNNs is primarily attributed to their message passing scheme, where each node updates its features by aggregating information of its neighbors (Corso et al., 2020; Xie et al., 2020). Nevertheless, with the remarkable growth in graph sizes (from millions to billions of nodes) over the past few years, GNNs have experienced substantial computational overheads during both model training and inference (Xu et al., 2019; You et al., 2020).

To address this inefficiency, current research in this area mainly follows two distinct lines of investigation – one focuses on *simplifying the graph structure*, while the other concentrates on *compressing the GNN model*. In the first line of study, previous literature has explored the utilization of sampling (Chen et al., 2018; Eden et al., 2018; Calandriello et al., 2018) and sparsification (Voudigari et al., 2016; Zheng et al., 2020; Li et al., 2020b) to reduce the computational overhead of GNNs. Compared to efforts in simplifying the graph structure, research on pruning or compressing GNNs (Tailor et al., 2020) has been relatively limited, mainly due to the inherent lower parameterization of GNNs compared to other fields such as computer vision (Wen et al., 2016; He et al., 2017).

---

[*]Yuxuan Liang and Jin Zeng are the corresponding authors, † denotes equal contributions.

Interestingly, a burgeoning subfield, inspired by the Lottery Ticket Hypothesis (Frankle & Carbin, 2018), is exploring the potential of jointly and iteratively pruning weights and adjacency matrices within GNNs. These pruned subnetworks and subgraphs, which can match the original baseline performance, are termed Graph Lottery Tickets (GLT) (Chen et al., 2021b; Hui et al., 2023). Specifically, UGS (Chen et al., 2021b) jointly prunes the weights and adjacency matrix and rewinds the weight at the regular iterations. TGLT (Hui et al., 2023) extends this concept and designs a new auxiliary loss function to guide the edges pruning for identifying GLT, to name just a few. Despite their remarkable performance, training GLT models is extremely challenging. Issues are arising in various aspects including predefined sparsity, network depth, and element pruning, potentially leading to the collapse of the entire model:

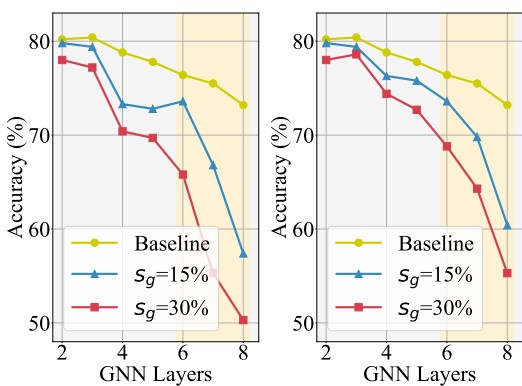

**Figure 1:** The performance of (*Left*) UGS and (*Right*) TGLT on the Cora dataset for GCN pruning at 2-8 layers, under graph sparsity levels of 0%, 15%, and 30%. See the definition of $s_g$ in Eq. G.

- **Inferior scalability in the context of deep GNNs.** Recent endeavors on GNN deepening (Li et al., 2019; 2020a; 2021), have already attested to the potential of deep GNNs. However, the off-the-shelf GLT algorithms demonstrate considerable sensitivity to model depth. As shown in Fig. 1, UGS and TGLT exhibit a significant performance decline when applied to deeper GCNs. For example, there is a performance degradation exceeding 10% from $6 \rightarrow 8$ GNN layers. This constrains the model's potential in the pursuit of ever-deeper GNNs.

- **Loss of elasticity in pruning process.** The majority of GLT methods prune weights and edges in an irreversible fashion, thereby rendering the pruned weights or edges irretrievable. Nevertheless, recent investigations (He et al., 2018; Mocanu et al., 2018; Evci et al., 2020) suggest that the significance of both edges and weights might dynamically evolve during the training process. As a result, the pruning methodologies previously employed in GLT exhibit a lack of elasticity.

- **Inflexibility of fixed pruning rates.** Prior GLT algorithms necessitate predetermined pruning ratios (e.g. 5% for graph and 20% for weight). Nonetheless, employing a fixed pruning configuration across all GNN layers lacks flexibility and resilience. Worse still, the trial-and-error selection process for these pre-defined parameters could introduce additional computational overhead.

In this paper, aiming to jointly overcome these crucial, intractable, and inherent hurdles, we propose an **A**daptive, **D**ynamic and **A**utomatic framework for identifying **G**raph **L**ottery **T**ickets, termed `AdaGLT`. Fig. 2 left demonstrates the overall workflow of `AdaGLT`. In contrast to previous GLT methods (see Fig. 2 right) which prune a fixed rate of weights or edges after certain epochs of pretraining, our method seamlessly performs sparsification and training within one epoch, thereby obtaining winning tickets at continuous sparsity levels. Our proposed algorithm is equipped with the following promising features, with strong experimental validation:

**Adaptive layer sparsification.** Recent studies have suggested that shallower layers, owing to their lower node similarity, should be assigned a more conservative pruning rate. As the network depth increases, the pruning rate should be correspondingly adjusted upwards to effectively alleviate issues like over-smoothing (Wang et al., 2023a). Classic GLT algorithms lack an effective adaptive mechanism, leading to suboptimal performance when applied to deep GNNs. `AdaGLT` is capable of learning layer-

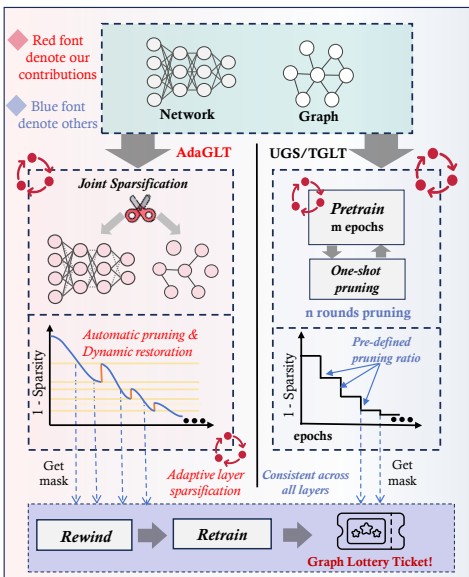

**Figure 2:** The overall workflow of our proposed `AdaGLT` compared with previous methods.

specific thresholds for GNN at different layers, enabling the model to acquire layer-adaptive sparse structures at each layer.

**Dynamic restoration.** In contrast to prior deterministic approaches (Chen et al., 2021b; Harn et al., 2022), `AdaGLT` can seamlessly integrate the pruning and training process, enabling a dynamic restoration of possibly mistakenly pruned edges or weights during subsequent training phases.

**Automated pruning scheduling.** `AdaGLT` eliminates the need for manually predefined pruning ratios. The sparsity of the graph and network grows progressively with the pruning threshold being automatically adjusted during the training process to discover the optimal sparse graph and network structure that best fits the downstream task. This ingredient is completely free of human labor of trial-and-error on pre-selected sparsity choices.

**Empirical Evidence.** `AdaGLT` has been empirically validated across diverse GNN architectures and tasks. The experimental results show that `AdaGLT` consistently surpasses UGS/TGLT across various graph/network sparsity configurations on all benchmark datasets (Cora, Citeseer, PubMed, and Open Graph Benchmark(OGB)). `AdaGLT` attains 23%-84% graph sparsity and 87%-99% weight sparsity, maintaining performance without compromise, exhibiting enhancements of 13%-30% in graph sparsity and 5%-10% in weight sparsity. In deep GNN scenarios, `AdaGLT` achieves a remarkable increase of up to 40% in graph sparsity and 80% in weight sparsity. This substantial demonstration underscores the immense potential of a fully automated GLT in real-world applications.

## 2 PRELIMINARY & RELATED WORK

**Notations.** We consider an undirected graph $\mathcal{G} = \{\mathcal{V}, \mathcal{E}\}$ where $\mathcal{V}$ and $\mathcal{E}$ are the sets of nodes and edges of $\mathcal{G}$ respectively. We use $\mathbf{X} \in \mathbb{R}^{N \times F}$ to denote the features matrix of $\mathcal{G}$, where $N = |\mathcal{V}|$ denotes the number of nodes on the graph. We use $\mathbf{x}_i = \mathbf{X}[i, \cdot]$ to represent the $F$-dimensional feature vector corresponding to node $v_i \in \mathcal{V}$. An adjacency matrix $\mathbf{A} \in \mathbb{R}^{N \times N}$ is employed to represent the connectivity between nodes, where $\mathbf{A}[i, j] = 1$ if $(v_i, v_j) \in \mathcal{E}$ else $\mathbf{A}[i, j] = 0$.

**Graph Neural Networks (GNNs).** GNNs mainly fall into spectral and spatial two categories. The spectral GNN is derived from spectral graph theory (Chung & Graham, 1997; McSherry, 2001; Defferrard et al., 2016; Levie et al., 2018), which leverages the eigenvalues and eigenvectors of the graph Laplacian matrix to encode and process graph information. Spatial GNN (Kipf & Welling, 2017a; Velickovic et al., 2017; Xu et al., 2019) excels in its flexibility and efficiency by aggregating neighborhood information. Among those, Graph Convolutional Networks (GCN) (Kipf & Welling, 2017a) can be deemed as the most popular model. Without sacrificing generality, we consider a GCN with two convolutional layers for node classification, whose formulation and objective function can be defined as follows:

$$\mathbf{Z} = \text{Softmax}\left(\hat{\mathbf{A}}\sigma(\hat{\mathbf{A}}\mathbf{X}\boldsymbol{\Theta}^{(0)})\boldsymbol{\Theta}^{(1)}\right), \quad \mathcal{L}(\mathcal{G}, \boldsymbol{\Theta}) = -\sum\nolimits_{v_i \in \mathcal{V}_l} y_i \log(z_i), \quad (1)$$

where $\mathbf{Z}$ denotes the model prediction, $\boldsymbol{\Theta} = (\boldsymbol{\Theta}^{(0)}, \boldsymbol{\Theta}^{(1)})$ denotes the weights, $\sigma(\cdot)$ denote the activation function, $\hat{\mathbf{A}} = \hat{\mathbf{D}}^{-\frac{1}{2}}(\mathbf{A} + \mathbf{I})\hat{\mathbf{D}}^{\frac{1}{2}}$ is the symmetric normalized adjacency matrix and $\hat{\mathbf{D}}$ is the degree mtrix of $\mathbf{A} + \mathbf{I}$. We minimize the cross-entropy loss $\mathcal{L}(\mathcal{G}, \boldsymbol{\Theta})$ over all labelled nodes $\mathcal{V}_l \subset \mathcal{V}$, where $y_i$ and $z_i$ represents the label and prediction of node $v_i$, respectively.

**Graph Sparsification & Lottery Tickets (GLT).** The Lottery Ticket Hypothesis (LTH) posits that a sparse and effective subnetwork can be extracted from a dense network through an iterative pruning approach (Frankle & Carbin, 2018; Frankle et al., 2019; Zhang et al., 2021). Initially observed within dense networks, LTH has garnered significant attention across diverse domains, including generative models (Chen et al., 2021c;a), speech recognition (Ding et al., 2021), large language models (Chen et al., 2020; Prasanna et al., 2020). Chen et al. (2021b) borrowed the concept from LTH and firstly unified *simplifying the graph strcuture* with *compressing the GNN model* in the GLT research line. Specifically, GLT is defined as a pair of core subgraphs and sparse sub-network, which can be jointly identified from the full graph and the original GNN model. Recent extensions of the GLT theory (You et al., 2022; Wang et al., 2023b) and new algorithms (Harn et al., 2022; Rahman & Azad, 2022; Liu et al., 2023; Wang et al., 2023c) have made GLT shine in the field of graph pruning research line. However, these algorithms lack sufficient flexibility in terms of sparsity, network depth, and element pruning, resulting in a lack of robustness in practical deployments.

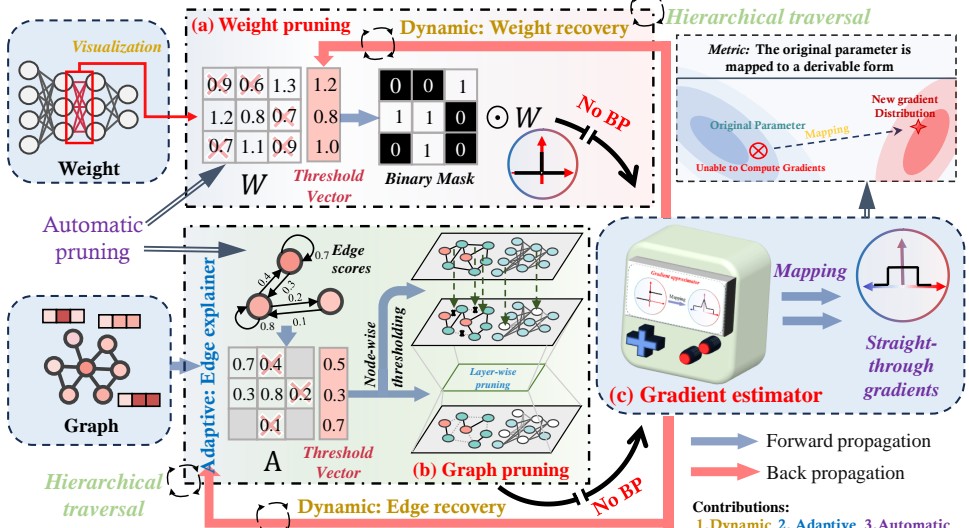

**Figure 3:** The framework of `AdaGLT`, in which we firstly allocate trainable threshold vectors to guide the weights and adjacency matrices pruning in different layers. (a) illustrates the weight pruning through row-wise thresholds. (b) showcases layer-adaptive pruning of the adjacency matrix. (c) elucidates how the gradient estimator renders the training process differentiable and enables the dynamic restoration of weights and edges.

## 3 METHODOLOGY

In this section, we proceed to provide an overarching depiction of the operational mechanics of `AdaGLT`, wherein adaptive, dynamic, and automated joint sparsification is executed for searching a GLT (Fig. 3). Taking a macro look, we undertake fine-grained weight sparsification, involving the creation of a binary mask through row-wise thresholding (Fig. 3 (a)). For the adjacency matrix, the edge explainer guides pruning by learning a soft mask (Fig. 3 (b)). Given the inherent non-differentiability of the binary mask during backpropagation, we employ a gradient estimator to simulate its gradients (Fig. 3 (c)). In the following parts, we will delve into the technical details of `AdaGLT`, focusing on its three core features and enhancement beyond mainstream GLT approaches.

### 3.1 AUTOMATED WEIGHT SPARSIFICATION

Existing GLT methods iteratively eliminate elements by choosing those with minimal magnitudes, and their lack of flexibility stems from the necessity of hyperparameter tuning. To rectify this drawback, AdaGLT dispenses with manually defined pruning rates and autonomously schedule the pruning process through the employment of **trainable threshold vectors**[1] (Liu et al., 2020; 2022; Zhou et al., 2021). Concretely, we allocate a set of trainable threshold vectors $\{t_\theta^{(0)}, t_\theta^{(1)}, \cdots, t_\theta^{(L-1)}\}$ to align with the weight set $\{\Theta^{(0)}, \Theta^{(1)}, \cdots, \Theta^{(L-1)}\}$ across $L$-layers GNN for sparsification. Within each layer, we calculate the binary mask $m_\theta$ via the magnitude comparison for weight pruning:

$$m_{\theta,ij} = \begin{cases} 1, & \text{if } |\Theta_{ij}| \geq t_{\theta,i} \\ 0, & \text{otherwise} \end{cases} \tag{2}$$

where $\Theta_{ij}$ and $m_{\theta,ij}$ is the element in the $i$-th row and $j$-th column of $\Theta$ and $m_\theta$, respectively; $t_{\theta,i}$ is the $i$-th element of the threshold vector $t_\theta$.

**Gradient Estimator.** However, it is not feasible to optimize these learnable thresholds using traditional gradient descent methods. The hard binary mask $m_\theta$ is derived through a comparison operation that prevents gradient backpropagation, rendering the thresholds untrainable. With this objective in mind, we calculate a soft differentiable mask using the tempered sigmoid function (Liu et al., 2022). Subsequently, we convert it into a hard binary mask as follows:

$$\tilde{m}_{\theta,ij} = \text{Sigmoid}\left(\tau \cdot (|\Theta_{ij}| - t_{\theta,i})\right), \ m_{\theta,ij} = \mathbb{1}[\tilde{m}_{\theta,ij} > 0.5], \tag{3}$$

where $\mathbb{1}[\tilde{m}_{\theta,ij} > 0.5]$ denotes an binary indicator that evaluates to 1 when $\tilde{m}_{\theta,ij} > 0.5$, and 0 otherwise. In order to approximate the hard mask, we leverage a temperature parameter $\tau$, wherein

---

[1]We discuss threshold scalar & matrix and compare their performances in Sec. 4.5 and Appendix B.

the Sigmoid function closely emulates the behavior of a step function when the temperature is elevated adequately. With the gradient Straight Through Estimator (STE)[2] (Bengio et al., 2013; Tian et al., 2021; Yang et al., 2023), we can compute the gradients of $t_\theta$ as follows:

$$\frac{\partial C}{\partial t_{\theta,i}} = -\sum_p \sum_q \frac{\partial C}{\partial \tilde{m}_{\theta,pq}} \delta_i^p \frac{\tau \exp\left(\tau(t_{\theta,p} - \Theta_{pq})\right)}{\left(1 + \exp\left(\tau(t_{\theta,p} - \Theta_{pq})\right)\right)^2}, \qquad (4)$$

where $C$ denotes the gradient of the cost in the context of back-propagation and $\delta_i^p$ evaluates to 1 when $i = p$, and 0 otherwise. Now that the calculation of pruning masks is differentiable, we proceed to calculate the pruned weights as follows:

$$\tilde{\mathbf{\Theta}}^{(l)} = \boldsymbol{m}_\theta^{(l)} \odot \mathbf{\Theta}^{(l)}, l = \{0, 1, \cdots, L-1\}, \qquad (5)$$

where $\odot$ denotes the element-wise multiplication and $\tilde{\mathbf{\Theta}}^{(l)}$ denotes the pruned weight at layer $l$. Upon obtaining the masked weights, they are employed in the subsequent convolutional operations. During the backward propagation phase, *gradients facilitated by STE concurrently update the weights of each layer and their corresponding threshold vectors.* Our weight sparsification process not only guides the weight updates towards enhanced performance but also aids in refining the threshold vectors to effectively unveil an optimal sparse structure within the network.

## 3.2 LAYER-ADAPTIVE ADJACENCY MATRIX PRUNING

Due to the discrete nature of adjacency matrix, we relax edge elements from binary variables to continuous variables and subsequently apply the same pruning strategy as weight pruning.

**Edge Explainer.** Conventional GLT methods often employ a trainable mask with its shape identical to that of the adjacency matrix, which may result in a quadratic increase in parameters with the size of the graph data (Sui et al., 2021). A promising and natural idea is to transform edge elements into a representation of importance scores, *i.e.*, relaxing edge elements from binary variables to continuous ones. With this in mind, we introduce the concept of edge explainer (Luo et al., 2020; Sui et al., 2022) into GNN pruning for the first time, ensuring interpretability during the pruning process while reducing unimportant edges. Specifically, we calculate the edge score according to the node features and node centrality as follows:

$$s_{ij} = \mathbb{1}_{(i,j)\in\mathcal{E}} \frac{\exp\left(g_\Psi(\mathbf{x}_i, \mathbf{x}_j)\right)/\omega}{\sum_{v_w \in \mathcal{N}(v_i)} \exp\left(g_\Psi(\mathbf{x}_i, \mathbf{x}_w)\right)/\omega}, \quad g_\Psi(\mathbf{x}_i, \mathbf{x}_j) = (W_Q\mathbf{x}_i)^T(W_K\mathbf{x}_j), \qquad (6)$$

where $s_{ij}$ is the edge score between $v_i$ and $v_j$, $g_\Psi$ is the edge explainer parameterized with $\Psi = \{W_Q, W_K\}$, $\mathcal{N}(v_i)$ denotes the 1-hop neighbors of $v_i$, and $\omega$ represents a temperature coefficient to control the importance scores. Once the weighted graph topology has been obtained, we proceed with the execution of layer-wise pruning. To achieve this objective, a trainable threshold vector is allocated for each layer's adjacency matrix, denoted collectively as $\{\boldsymbol{t}_A^{(0)}, \boldsymbol{t}_A^{(1)}, \cdots, \boldsymbol{t}_A^{(L-1)}\} \in \mathbb{R}^N$, mirroring the analogous allocation of threshold vectors for weights across each layer.

**Layer-adaptive Pruning.** Going beyond element pruning like weight sparsification, edge pruning seeks to obtain an optimal graph substructure with increasing sparsity across layers (especially in deep GNN circumstances). Consequently, we compute the pruning mask for each layer's adjacency matrix in an iterative manner:

$$\boldsymbol{m}_{A,ij}^{(l)} = \mathbb{1}[s_{ij} < \boldsymbol{t}_{A,i}^{(l)}] \prod_{k=0}^{l-1} \boldsymbol{m}_{A,ij}^{(k)}, \; l = \{0, 1, \cdots, L-1\}, \qquad (7)$$

where $\boldsymbol{m}_A^{(l)} \in \mathbb{R}^{N \times N}$ denotes the graph mask at layer $l$. The product term $\prod_{k=0}^{l-1} \boldsymbol{m}_{A,ij}^{(k)}$ ensures that edges pruned in earlier layers are also retained as pruned in the $l$-th layer, thus compelling the searched graph substructure to exhibit progressively escalating sparsity across layers. Similarly, we utilize a gradient estimation approach akin to that delineated in Eq. 3 and 4 to address the non-differentiability inherent in binary pruning masks. Ultimately, we attain distinct and layer-wise sparse graph topological structures, denoted as $\tilde{\mathbf{A}}^{(l)} = \boldsymbol{m}_A^{(l)} \odot \mathbf{A}$ and $l = \{0, 1, \cdots, L-1\}$.

## 3.3 A UNIFIED AND DYNAMIC OPTIMIZATION

**Sparse Regularization.** In pursuit of achieving the desired levels of sparsity for both weights and adjacency matrices, we seek to obtain threshold vectors with higher values. To this end, it becomes

---

[2]We discuss alternative estimators and compare their performance in Appendix K.

necessary to impose sparsity-inducing constraints on $\boldsymbol{t}_\theta$ and $\boldsymbol{t}_A$ to penalize smaller threshold values. Specifically, for any given threshold vector $\boldsymbol{t} \in \mathbb{R}^N$, its associated penalty term is given by $\mathcal{R}(\boldsymbol{t}) = \eta \sum_{i=1}^{N} \exp(-\boldsymbol{t}_i)$ and $\eta$ is the regularization coefficient. We unify the sparsity penalties for the threshold vectors of weights and adjacency matrices as $\mathcal{L}_s = \frac{1}{L} \sum_{l=0}^{L-1} (\mathcal{R}(\boldsymbol{t}_\theta^{(l)}) + \mathcal{R}(\boldsymbol{t}_A^{(l)}))$.

**Objective function.** The original fully-connected layers and the underlying graph topology have now been replaced with finely-grained masked weights and progressively sparse adjacency matrices. We can proceed to directly train a jointly sparsed GNN using the backpropagation algorithm. In our approach, the network weights $\boldsymbol{\Theta}$, edge explainer $\Psi$, as well as the weight threshold vector $\boldsymbol{t}_\theta$ and adjacency threshold vector $\boldsymbol{t}_A$ are concurrently trained in an end-to-end way:

$$\mathcal{L}_{\texttt{AdaGLT}}(\boldsymbol{\Theta}, \Psi, \boldsymbol{t}_\theta, \boldsymbol{t}_A) = \mathcal{L}(\{\boldsymbol{m}_\text{A} \odot \mathbf{A}, \mathbf{X}\}, \boldsymbol{m}_\theta \odot \boldsymbol{\Theta}) + \mathcal{L}_s, \tag{8}$$

$$\boldsymbol{m}_\text{A}^*, \boldsymbol{m}_\theta^* = g\left(\underset{\boldsymbol{\Theta}, \Psi, \boldsymbol{t}_\theta, \boldsymbol{t}_A}{\arg\min} \mathcal{L}_{\texttt{AdaGLT}}(\boldsymbol{\Theta}, \Psi, \boldsymbol{t}_\theta, \boldsymbol{t}_A)\right), \tag{9}$$

where $\mathcal{L}(\cdot)$ denotes the cross-entropy loss, and $g(\cdot)$ represents inferring the optimal masks using parameters that minimize the loss. To facilitate reading, we show the algorithm in Algo. 1.

**Dynamic Pruning and Restoration.** From Sec. 3.1 to 3.3, we systemacially haved elaborated on how `AdaGLT` helps to find adaptive and automated wining tickets. The adaptiveness and automation of the GLT algorithms, are at present obscured by one cloud (*dynamic sparsification*). In this part, we will explain how a pruned element can be restored through gradient updates. Specifically, considering a layer's adjacency matrix or weight (unified as $Q \in \mathbb{R}^{N \times M}$), with its threshold vector denoted as $\boldsymbol{t} \in \mathbb{R}^N$, the resulting pruning mask is represented as $\boldsymbol{m} \in \mathbb{R}^{N \times M}$. At the $n$-th epoch, if the value at $Q_{ij}$ (i.e. edge score or weight value) satisfies $Q_{ij} < \boldsymbol{t}_i$, then $\boldsymbol{m}_{ij} = 0$. Although this element is pruned in this epoch, during the gradient backpropagation process, it can still be dynamically revived by the gradient, subject to the following conditions:

$$|Q_{ij}| - \boldsymbol{t}_i > \alpha \left(\frac{\partial C}{\partial Q_{ij}} - \frac{\partial C}{\partial \boldsymbol{t}_i}\right) = \alpha \sum_p \sum_q \frac{\partial C}{\partial Q_{pq}} (\delta_q^j + 1)\delta_i^p \frac{\tau \exp(\tau(\boldsymbol{t}_p - Q_{pq}))}{(1 + \exp(\tau(\boldsymbol{t}_p - Q_{pq})))^2}, \tag{10}$$

where $\alpha$ denotes the learning rate. It can be observed that the restoration of an element depends on the joint update of $Q_{ij}$ and $\boldsymbol{t}_i$. During the collaborative optimization between element values and thresholds, the optimal sparse structure for both the graph and weights is dynamically obtained.

## 3.4 MODEL SUMMARY & THEORETICAL DICUSSIONS

After introducing the model details, we further offer a theoretical guarantee for our layer-adaptive sparsification grounded in the well-established Graph Neural Tangent Kernel (GNTK) theory, which is pertinent to deep GNNs (Huang et al., 2021). The GNTK can be formally defined as:

$$\mathbf{K}_t(X, X) = \sum_{l=1}^{L} \nabla_\theta h_t^{(l)}(X) \nabla_\theta h_t^{(l)}(X)^T \in \mathbb{R}^{N \times N}, \tag{11}$$

where $h^{(l)}$ denotes the embedding representation at layer $l$. It is widely recognized that the GNTK plays a central role in optimizing infinitely-wide GNNs, as documented in various studies (Jacot et al., 2018; Huang et al., 2021). Specifically, a positive minimum eigenvalue is indicative of the successful training of the corresponding GNN through gradient descent. Conversely, the inability to achieve a minimal training error suggests the presence of a zero or negative minimum eigenvalue. The propagation of GNTK with respect to graph convolution can be expressed as follows. With a more compact expression when the tangent kernel is bacterized, we have $\mathbf{k}^{(l)} = \mathbf{G}^{(l)} \mathbf{k}^{(l-1)}$, where $\mathbf{k} \in \mathbb{R}^{N^2}$ and $\mathbf{G}^{(l)} \in \mathbb{R}^{N^2 \times N^2}$ is the operation matrix for GNTK propagation. By looking at $\mathbf{k}^{(l)}$ with $l$ tends to infinity, we scrutinize the efficacy of layer-adaptive sparsification in mitigating the prevalent issue of over-smoothing.

**Theorem 1.** *Denote smallest singluar value of $\mathbf{G}^{(l)}$ by $\sigma_0^{(l)}$. Suppose that operation matrix $\mathbf{G}^{(l)}$ has the same singlular vectors, and $\sigma_0^{(l)}$ satifies $\sigma_0^{(l)} = 1 - \alpha^l$, where $0 < \alpha < 1$. Then, the smallest eigenvalue of GNTK is greater than zero as the depth of GNN tends to infinity, i.e., $\lim_{l \to \infty} \lambda_0(\mathbf{K}^{(l)}) > 0$.*

Theorem 1 states that as the depth tends to be infinity, the GNTK can still have a positive smallest eigenvalue, which implies that the corresponding GNN can be trained successfully. This result is opposite to the case without graph sparsification (Huang et al., 2021), demonstrating the effectiveness of layer-wise sparsification. We have provided detailed proofs in Appendix E.

# 4 EXPERIMENTS

In this section, we conduct extensive experiments to answer the following research questions: **RQ1:** How effective is our proposed `AdaGLT` algorithm in searching graph lottery tickets? **RQ2:** What is the scalability of `AdaGLT` on deeper GNNs? **RQ3:** Can `AdaGLT` scale up to large-scale datasets?

## 4.1 EXPERIMENTAL SETUPS

**Datasets.** To comprehensively evaluate `AdaGLT` across diverse datasets and tasks, we opt for Cora, Citeseer, and PubMed (Kipf & Welling, 2017b) for node classification. For larger graphs, we choose Ogbn-Arxiv/Proteins/Products (Hu et al., 2020) for node classification and Ogbl-Collab for link prediction. Detailed dataset statistics can be found in Appendix F.

**Backbones & Parameter Settings.** We conduct comparative analyses between `AdaGLT` and conventional GLT (Chen et al., 2021b) and TGLT (Hui et al., 2023) in all available scenarios. For Cora, Citeseer, and PubMed, we employ GCN (Kipf & Welling, 2017b), GIN (Xu et al., 2019), and GAT (Veličković et al., 2018) as backbones. For Ogbn-Arxiv/Proteins and Ogbl-Collab, we employ DeeperGCN (Li et al., 2020a) as the backbone, while for Ogbn-Products, we opt for Cluster-GCN (Chiang et al., 2019). More experimental details are specified in Appendix G.

## 4.2 CAN `AdaGLT` FIND GRAPH LOTTERY TICKETS? (RQ1)

To answer RQ1, we conduct a comparative analysis between `AdaGLT` and existing methodologies, including UGS, TGLT, and random pruning, on the node classification tasks. The results on Citeseer and PubMed are presented in Fig. 4, and those on Cora are displayed in Fig. 9. From the experimental results, we can make the following observations (**Obs**):

**Obs 1. `AdaGLT` consistently outperforms TGLT, UGS, and random pruning.** Across Cora, Citeseer, and PubMed datasets, `AdaGLT` successfully identifies graph lottery tickets with graph sparsity

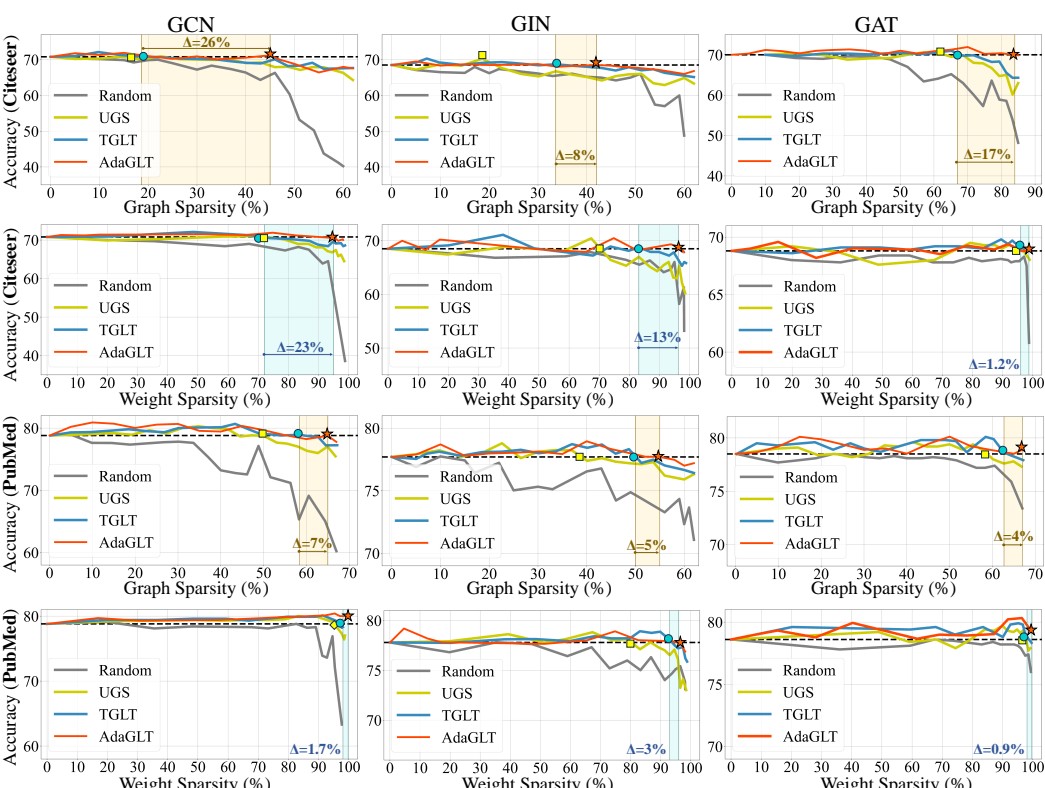

**Figure 4:** Results of node classification over Citeseer/PubMed with GCN/GIN/GAT backbones. Black dash lines represent the baseline performance. Marker ■, ● and ★ indicates the last GLT that reaches higher accuracy than the original model in the sparsification process of UGS, TGLT, and `AdaGLT`, respectively. Δ quantifies the percentage by which our AdaGLT method outperforms the state-of-the-art GLT methods.

**Table 1:** The performance comparison between TGLT and `AdaGLT` in discovering GLTs on GAT backbone across various graph sparsity settings (10% → 60%) and GNN layer configurations (4 → 16 layers). Cells highlighted in red and blue correspond to winning tickets found by TGLT and `AdaGLT`, respectively.

| Graph Sparsity | Method | Cora | | | | Citeseer | | | | PubMed | | | |
|---|---|---|---|---|---|---|---|---|---|---|---|---|---|
| | | 4 | 8 | 12 | 16 | 4 | 8 | 12 | 16 | 4 | 8 | 12 | 16 |
| 0% | Baseline | 78.20 | 78.08 | 76.12 | 75.53 | 69.82 | 67.50 | 67.40 | 67.56 | 78.10 | 76.82 | 76.30 | 76.81 |
| 10% | TGLT[2023] | 78.79 | 79.39 | 66.32 | 60.59 | 70.11 | 67.79 | 60.90 | 61.88 | 78.62 | 77.02 | 76.79 | 61.22 |
| | AdaGLT | 79.82 | 78.77 | 77.24 | 74.28 | 69.88 | 67.67 | 68.11 | 67.99 | 78.43 | 78.19 | 77.76 | 75.29 |
| 20% | TGLT[2023] | 73.70 | 78.80 | 40.79 | 36.77 | 69.86 | 56.88 | 58.10 | 53.33 | 78.30 | 69.88 | 68.49 | 59.27 |
| | AdaGLT | 78.64 | 78.22 | 76.88 | 75.60 | 70.03 | 68.30 | 67.24 | 68.03 | 78.44 | 78.23 | 76.80 | 73.03 |
| 30% | TGLT[2023] | 70.84 | 76.22 | 44.83 | 42.88 | 66.21 | 55.45 | 53.42 | 47.99 | 75.07 | 65.98 | 59.42 | 58.60 |
| | AdaGLT | 78.39 | 78.48 | 76.70 | 72.75 | 69.91 | 67.39 | 67.88 | 67.90 | 79.24 | 77.11 | 74.99 | 70.47 |
| 40% | TGLT[2023] | 72.60 | 72.41 | 41.88 | 39.77 | 60.82 | 47.80 | 50.88 | 49.21 | 68.80 | 69.72 | 60.77 | 55.42 |
| | AdaGLT | 77.33 | 74.91 | 74.30 | 69.98 | 69.83 | 67.50 | 67.58 | 65.33 | 78.33 | 76.88 | 74.39 | 71.26 |
| 50% | TGLT[2023] | 72.17 | 72.91 | 40.07 | 45.80 | 61.31 | 48.77 | 51.62 | 45.90 | 65.24 | 68.74 | 63.22 | 60.33 |
| | AdaGLT | 77.11 | 75.07 | 73.18 | 72.60 | 69.97 | 64.10 | 65.31 | 64.88 | 77.42 | 76.90 | 75.30 | 74.95 |
| 60% | TGLT[2023] | 70.60 | 66.50 | 37.74 | 37.26 | 52.37 | 54.40 | 50.85 | 47.25 | 62.98 | 66.07 | 63.66 | 60.79 |
| | AdaGLT | 70.97 | 75.00 | 72.29 | 71.88 | 68.13 | 64.98 | 64.34 | 60.97 | 75.82 | 73.23 | 69.47 | 69.01 |

from 23% to 84% and weight sparsity from 87% to 99%, all while sustaining performance. Notably, `AdaGLT` can find winning tickets on GCN for Citeseer, whose graph and weight sparsity separately reach 44% and 95%, significantly surpassing TGLT's 26% and 23%.

**Obs 2. `AdaGLT` shows particular efficacy in uncovering sparse graphs.** While weight pruning has been well developed, attaining GLTs with sufficient graph sparsity remains challenging (Wang et al., 2022). PubMed is often considered more robust against graph pruning, whereas the smaller-sized Cora and Citeseer are deemed more sensitive (Chen et al., 2021b; Wang et al., 2023b). It is worth noting that `AdaGLT` outperforms TGLT in terms of graph sparsity, achieving improvements of 26% and 18% on Citeseer and Cora, respectively. Particularly remarkable is our method's ability to prune 80% of edges when applied to GAT on Citeseer without any performance deterioration.

**Obs 3. The GNN backbone intricately affects graph sparsity attainment.** While `AdaGLT` consistently identifies GLTs with weight sparsity over 90%, its performance with graph sparsity relies more on the specific GNN structure. Concretely, in GAT, GLTs are found with graph sparsity over 65%, whereas those in GIN do not exceed 55%. We attribute this to GAT's attention-based aggregation, which adapts during the edge pruning process. However, as for GIN, its aggregation of outputs from all layers amplifies the information loss caused by edge pruning.

### 4.3 Can `AdaGLT` handle deeper GNNs? (RQ2)

Tab. 1 and Tab. 6 to 10 present a comparison between TGLT and `AdaGLT` on GCN/ResGCN/GAT backbones, with the number of GNN layers increasing from 4 to 16, graph sparsity from 10% to 60%, and weight sparsity from 10% to 90%. Our findings and insights are summarized below:

**Obs 4. The increase in the number of GNN layers poses challenges for identifying GLTs.** In Tab. 1, both TGLT and `AdaGLT` exhibit a reduction in the number of lottery tickets as # of layers increases. Under 4-layer settings, the sparsity of the uncovered tickets is roughly similar to that on 2-layer GNNs. However, on 16-layer GNNs, both methods fail to identify extremely sparse GLTs.

**Obs 5. `AdaGLT` excels at discovering GLTs in deep GNNs.** In Tab. 6 to 10, in comparison to the excellent performance observed in 2-layer GNNs, TGLT suffers setbacks in identifying GLTs. Even on the GCN backbone where it performs at its best, TGLT can barely identify tickets with graph sparsity of 20% and weight sparsity of 50%. In contrast, `AdaGLT` can identify GLTs with graph and weight sparsity exceeding 40% and 70% on Citeseer and PubMed across all settings.

### 4.4 Can `AdaGLT` scale up to large-scale dataset? (RQ3)

Fig. 5 and Fig. 10 to 13 illustrate the performance of `AdaGLT` on DeeperGCN across depths ranging from 4 → 28 layers, evaluated on Ogbn-Arxiv/Proteins and Ogbl-Collab. We can list observations:

**Obs 6. `AdaGLT` is capable of learning layer-wise sparse graph structures.** As depicted in Fig. 6, the GLTs discovered by `AdaGLT` exhibit a progressively sparser graph topology across layers, with their graph sparsity increasing from 16% to as high as 70% across layers.

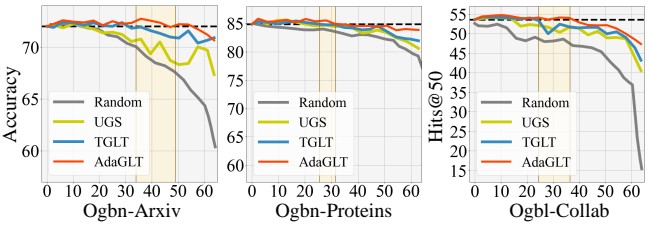 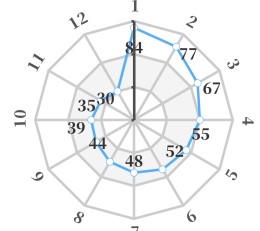

**Figure 5:** The performance of `AdaGLT` compared with UGS, TGLT and random pruning on 12-layer DeeperGCN with Ogbn-Arxiv/Proteins and Ogbl-Collab. Black dashed lines represent the baseline performance.

**Figure 6:** The percentage of remaining edges at each layer of a 12-layer DeeperGCN after applying `AdaGLT`.

**Obs 7.** `AdaGLT` **can scale up to large graphs.** As depicted in Fig. 5 and Fig. 10 to 13, `AdaGLT` consistently outperforms TGLT and UGS. In line with the findings from **Obs 6**, with the increase in the number of GNN layers, TGLT and UGS struggle to find high graph sparsity graph lottery tickets. In contrast, `AdaGLT` exhibits greater robustness to the number of layers, surpassing TGLT by 21%, 7%, and 11% in graph sparsity with the 28-layer setting, respectively.

## 4.5 ABLATION STUDY

**Effects of Threshold Level.** In our comparison across different threshold levels on Cora, Citeseer, and PubMed, as depicted in Fig. 7 and 8, we have made several noteworthy observations. Underline{First}, on small graphs like Cora, the threshold matrix consistently outperforms the threshold scalar/vector (with GLTs of over 40% graph sparsity and 99% weight sparsity). This superiority can be attributed to its ability to adjust the threshold element-wisely; Second, when applied to larger graphs like Citeseer and PubMed, the threshold vector exhibits better performance, especially in sparsifying the graph. We believe this is because the threshold vector strikes a balance between precision and parameter efficiency. Unlike the threshold scalar, it does not uniformly apply a single threshold across the entire matrix. Additionally, it avoids introducing excessive parameters like the threshold matrix during training. In summary, we adopt the threshold vector for all experiments.

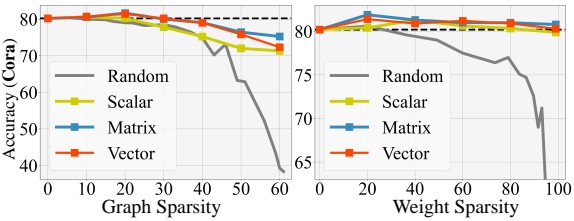

**Figure 7:** Ablation study on different threshold levels (i.e. threshold scalar, vector, and matrix) with GCN on Cora dataset. Black dash lines represent the baseline performance.

| Settings | | Graph Sparsity | | | | |
|---|---|---|---|---|---|---|
| Dataset | Estimator | 20% | 30% | 40% | 50% | 60% |
| Cora +GCN | STE | 80.6 | 79.8 | 80.0 | 78.7 | 74.5 |
| | LTE | 80.6 | 80.1 | 79.2 | 78.9 | 74.4 |
| | SR-STE | 80.4 | 78.8 | 78.3 | 77.7 | 74.4 |
| Citeseer +GIN | STE | 69.7 | 69.6 | 68.6 | 68.4 | 67.2 |
| | LTE | 69.8 | 69.4 | 68.5 | 68.2 | 67.0 |
| | SR-STE | 69.5 | 69.4 | 68.0 | 67.8 | 66.7 |

**Table 2:** Ablation study on different gradient estimators over different datasets and backbones. Cells highlighted in blue represent the performance of found GLTs.

**Effects of Gradient Estimator.** To assess the sensitivity of `AdaGLT` to various gradient estimators, we compared STE with two other popular gradient estimators: Long-Tailed Estimator (LTE) (Xu & Cheung, 2019; Liu et al., 2020) and SR-STE (Zhou et al., 2021). As depicted in Tab. 2, the performance of different estimators is generally similar, demonstrating that `AdaGLT` is insensitive to the choice of estimators. More details can be found in Appendix K.

## 5 CONCLUSION

In this paper, we propose an adaptive, dynamic and automatic framework for identifying graph lottery tickets (`AdaGLT`) that unifies graph and weight sparsification in a dynamic and automated workflow and is capable of winning layer-adaptive tickets. We provide theoretical substantiation for this layer-adaptive sparsification paradigm in the context of deep GNNs. To verify the effectiveness of `AdaGLT`, we conduct extensive experiments and ablations across different graph benchmarks, various backbones and depth settings of GNNs. Our experiments consistently demonstrate its superiority over existing GLT methods. These findings shed light on automating the process of graph lottery ticket discovery and expanding the generality of such tickets.

## 6 ACKNOWLEDGEMENT

This work was supported in part by the National Natural Science Foundation of China under Grant 62201389, the Shanghai Rising-Star Program under Grant 22YF1451200, the Guangzhou-HKUST(GZ) Joint Funding Program (No. 2024A03J0620) and the Tongji University Undergraduate Innovation and Entrepreneurship Program (No. 202310247097 ).

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

# A   NOTATIONS

**Table 3:** The notations that are commonly used in Methodology (Sec. 3).

| Notation | Definition |
|---|---|
| $\mathcal{G} = \{\mathcal{V}, \mathcal{E}\}$ | Input graph |
| $\mathbf{A}$ | Input adjacency matrix |
| $\mathbf{X}$ | Input features |
| $\mathbf{\Theta}$ | $\mathbf{\Theta} = \left\{ \mathbf{\Theta}^{(0)}, \mathbf{\Theta}^{(1)}, \cdots, \mathbf{\Theta}^{(L-1)} \right\}$ represent the weight matrices of GNN |
| $\boldsymbol{t}_\theta$ | $\boldsymbol{t}_\theta = \left\{ \boldsymbol{t}_\theta^{(0)}, \boldsymbol{t}_\theta^{(1)}, \cdots, \boldsymbol{t}_\theta^{(L-1)} \right\}$ represent threshold vectors for weight sparsification |
| $\boldsymbol{t}_{\mathrm{A}}$ | $\boldsymbol{t}_{\mathrm{A}} = \left\{ \boldsymbol{t}_{\mathrm{A}}^{(0)}, \boldsymbol{t}_{\mathrm{A}}^{(1)}, \cdots, \boldsymbol{t}_{\mathrm{A}}^{(L-1)} \right\}$ represent threshold vectors for graph sparsification |
| $\boldsymbol{m}_\theta$ | $\boldsymbol{m}_\theta^{(l)} = \left\{ \boldsymbol{m}_\theta^{(0)}, \boldsymbol{m}_\theta^{(1)}, \cdots, \boldsymbol{m}_\theta^{(L-1)} \right\}$ represent the weight pruning masks |
| $\boldsymbol{m}_{\mathrm{A}}$ | $\boldsymbol{m}_{\mathrm{A}}^{(l)} = \left\{ \boldsymbol{m}_{\mathrm{A}}^{(0)}, \boldsymbol{m}_{\mathrm{A}}^{(1)}, \cdots, \boldsymbol{m}_{\mathrm{A}}^{(L-1)} \right\}$ represent the graph pruning masks |
| $s_{ij}$ | Edge score between $v_i$ and $v_j$ |
| $\Psi$ | $\Psi = \{W_K, W_V\}$ denotes parameters of edge explainer |
| $\mathbf{H}^{(l)}$ | Embedding representation after $l$-th GNN embedding layer |
| $\tilde{\mathbf{\Theta}}^{(l)}$ | Sparsified weight matrix at $l$-th GNN layer |
| $\tilde{\mathbf{A}}^{(l)}$ | Sparsified adjacency marix at $l$-th GNN layer |

# B   DETAILS ON THRESHOLD LEVEL

In addition to the threshold vector, we can also employ a threshold scalar or threshold matrix to learn adaptive thresholds for each weight matrix or adjacency matrix (Liu et al., 2020). Specifically, considering a layer's adjacency matrix or weight (unified as $Q \in \mathbb{R}^{N \times M}$), we define its threshold scalar and threshold matrix as $\boldsymbol{t}$ and $\mathbf{T} \in \mathbb{R}^{N \times M}$, respectively. The calculation of the binary pruning mask is as follows:

$$m_{ij} = \begin{cases} \mathbb{1}[|Q_{ij}| > \boldsymbol{t}], & \text{for threshold scalar} \\ \mathbb{1}[|Q_{ij}| > \mathbf{T}_{ij}], & \text{for threshold matrix} \end{cases} \tag{12}$$

where $m$ denotes the resulting binary pruning mask. We conduct comparative experiments on Cora, Citeseer, and Pubmed datasets to assess the reliability and effectiveness of thresholds at various levels.

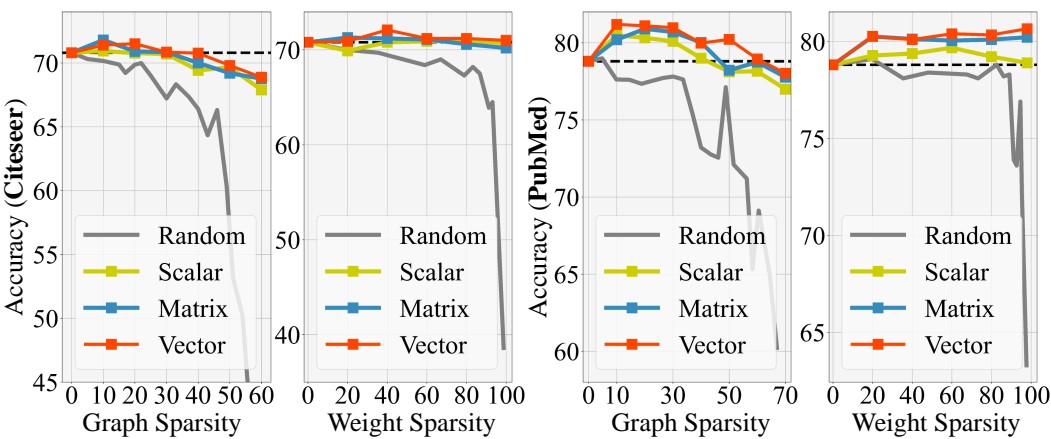

**Figure 8:** Ablation study on different threshold levels (i.e. threshold scalar, vector, and matrix) with GCN on Citeseer/PubMed dataset. Black dash lines represent the baseline performance.

## C    ALGORITHM FRAMEWORK OF `AdaGLT`

In this section, we conclude the overall algorithm framework of `AdaGLT` in Algo. 1. During each epoch, we first compute the edge scores with the edge explainer (Eq. 3.2) and calculate the pruning masks for weight and adjacency matrix in a layer-wise manner (Eq. 2 & 7). After that, we forward the network with sparsed weights and graphs and updates parameter through gradient estimators. It is important to note that, unlike conventional network training, our objective is not to attain the optimal network parameters but rather to achieve the *optimal sparse structure* (i.e. $\boldsymbol{m}_A^*, \boldsymbol{m}_\theta^*$). We draw the *optimal sparse structure* from pruning masks with sparsity levels within the desired range and with the best performance on the validation set.

---

**Algorithm 1:** Algorithm workflow of `AdaGLT`

---

**Input** : $\mathcal{G} = (\mathbf{A}, \mathbf{X})$, GNN model $f(\mathcal{G}, \boldsymbol{\Theta}_0)$, GNN's initialization $\boldsymbol{\Theta}_0$, threshold vectors $\mathbf{t}_\theta$ and $\mathbf{t}_A$, step size $\eta$, target graph sparsity $s_g$ and weight sparsity $s_\theta$, sparsity interval width $\epsilon$

**Output:** GLT $f\left(\{\boldsymbol{m}_A \odot \mathbf{A}, \mathbf{X}\}, \boldsymbol{m}_\theta \odot \boldsymbol{\Theta}\right)$

1 **for** iteration $i \leftarrow 1$ **to** $N$ **do**

2     Compute edge scores $s_{ij} = \mathbb{1}_{(i,j)\in\mathcal{E}} \frac{\exp\left(g_\Psi(\mathbf{x}_i, \mathbf{x}_j)\right)/\omega}{\sum_{v_w \in \mathcal{N}(v_i)} \exp\left(g_\Psi(\mathbf{x}_i, \mathbf{x}_w)\right)/\omega}$ (Eq. 3.2)

3     **for** layer $l \leftarrow 1$ **to** $L$;                  ▷ Dynamic & Automated Pruning

4     **do**

5        $\boldsymbol{m}_{\theta,ij}^{(l)} \leftarrow \mathbb{1}[|\boldsymbol{\Theta}_{ij}| - \boldsymbol{t}_{\theta,i}]$ (Eq. 2)

6        $\boldsymbol{m}_{A,ij}^{(l)} = \mathbb{1}[s_{ij} < t_{A,i}^{(l)}] \prod_{k=0}^{l-1} \boldsymbol{m}_{A,ij}^{(k)}$ (Eq. 7) ;            ▷ Layer-adaptive

7     **end**

8     Forward $f(\cdot, \boldsymbol{m}_\theta \odot \boldsymbol{\Theta})$ with $\mathcal{G} = \{\boldsymbol{m}_A \odot \mathbf{A}, \mathbf{X}\}$ to compute the loss $\mathcal{L}_{\texttt{AdaGLT}}$ in Eq. 3.3.

9     Backpropagate to update $\boldsymbol{\Theta}_{i+1} \leftarrow \eta \nabla_{\boldsymbol{\Theta}_i} \mathcal{L}_{\texttt{AdaGLT}}, \Psi_{i+1} \leftarrow \eta \nabla_{\Psi_i} \mathcal{L}_{\texttt{AdaGLT}}$

10     Update $\boldsymbol{t}_\theta$ and $\boldsymbol{t}_A$ with STE according to Eq. 2. ;        ▷ Dynamic restoration

11     Compute current graph and weight sparsity $c_g$ and $c_\theta$.

12     **if** $|c_g - s_g| \leq \epsilon$ and $|c_\theta - s_\theta| \leq \epsilon$ **then**

13        Update the optimal mask $\boldsymbol{m}_A^*, \boldsymbol{m}_\theta^* \leftarrow \boldsymbol{m}_A, \boldsymbol{m}_\theta$ (if with higher validation score)

14 **end**

15 Rewind GNN's weights to $\boldsymbol{\Theta}_0$.

16 Retrain the model with fixed $\boldsymbol{m}_A^*$ and $\boldsymbol{m}_\theta^*$.

---

## D    COMPLEXITY ANALYSIS

Following Chen et al. (2021b), we exhibit the complexity of `AdaGLT`. The inference time complexity of UGS is defined as $\mathcal{O}\left(L \times ||\boldsymbol{m}_A \odot \mathbf{A}||_0 \times D + L \times ||\boldsymbol{m}_\theta||_0 \times |\mathcal{V}| \times F^2\right)$, where $L$ is the number of layers, $||\boldsymbol{m}_A \odot \mathbf{A}||_0$ is the number of remaining edges in sparse graph, $D$ is the dimension of feature and $|\mathcal{V}|$ is the number of nodes. The inference time complexity of `AdaGLT` can be defined as $\mathcal{O}\left(L \times \sum_l \left\|\boldsymbol{m}_A^{(l)} \odot \mathbf{A}\right\|_0 \times D + E \times F + L \times ||\boldsymbol{m}_\theta||_0 \times |\mathcal{V}| \times F^2\right)$, where $\sum_l \left\|\boldsymbol{m}_A^{(l)} \odot \mathbf{A}\right\|_0$ represents the remaining edges across all layers and $E \times F$ denotes the complexity of calculating edge scores according to Eq. 3.2. It is worth mentioning that we solely compute edge scores and their similarities for edges $e_{ij}$ where $(i, j) \in \mathcal{E}$, thus avoiding the $\mathcal{O}(N^2)$ complexity associated with all-pair similarity computations.

## E    PROOF OF THEOREM 1

*Proof of Theorem 1.* We first study the propagation of GNTK with respect to graph convolution. The expression of graph convolution is governed as follows:

$$h^{(l)}(u) = \frac{1}{|\mathcal{N}(u)| + 1} \sum_{v \in \mathcal{N}(u) \cup u} h^{(l-1)}(v), \tag{13}$$

where $u, v$ denote the node index of the graph, and $\mathcal{N}(u) \cup u$ is the union of node $u$ and its neighbors. Equation (13) reveals the node feature aggregation operation among its neighborhood according to a GCN variant (Hamilton et al., 2017). Based on the definition of NTK (11), we recursively formulate the propagation of GNTK in the infinite-width limit. As information propagation in a GCN is built on aggregation (13), the corresponding formulas of GNTK are expressed as follows,

$$K^{(l)}(u, u') = \frac{1}{|\mathcal{N}(u)| + 1} \frac{1}{|\mathcal{N}(u')| + 1} \sum_{v \in \mathcal{N}(u) \cup u} \sum_{v' \in \mathcal{N}(u') \cup u'} K^{(l-1)}(v, v')$$

In order to facilitate calculation, we rewrite the above equation in the format of a matrix,

$$\mathbf{k}^{(l)} = \mathbf{G}^{(l)} \mathbf{k}^{(l-1)}$$

where $\mathbf{k}^{(l)} \in \mathbb{R}^{n^2 \times 1}$, is the result of GNTK being vectorized. Thus, the matrix operation $\mathbf{G}^{(l)} \in \mathbb{R}^{n^2 \times n^2}$.

Now we consider the impact of graph sparsification on the operation matrix $\mathbf{G}^{(l)}$. Different from what has been studied in (Huang et al., 2021) the operation matrix $\mathbf{G}^{(l)}$ is irreducible and aperiodic, with a stationary distribution vector. Here we consider the graph sparsification technique. In this case, the operation matrix is no longer a transition matrix that corresponds to a Markov process.

To study the limit behavior of the smallest eigenvalue of the GNTK, we make an SVD decomposition of the operation matrix in each layer as follows:

$$\mathbf{G}^{(l)} = \mathbf{U}^{(l)} \mathbf{\Sigma}^{(l)} \mathbf{V}^{(l)^\top}$$

where $\mathbf{U}^{(l)}$ and $\mathbf{V}^{(l)}$ are both orthogonal matrix. Because the operation matrix $\mathbf{G}^{(l)}$ is a symmetric matrix, we know that $\mathbf{U}^{(l)} = \mathbf{V}^{(l)}$. Based on our assumption that that operation matrix $\mathbf{G}^{(l)}$, for every layer, has the same singular vectors, which means that:

$$\mathbf{U}^{(l)} = \mathbf{U}^{(l-1)} = \cdots = \mathbf{U}^{(1)}$$

Then we can obtain the expression for GNTK at the final layer:

$$\mathbf{k}^{(l)} = \mathbf{U} \mathbf{\Sigma}^{(l)} \mathbf{V}^\top \mathbf{U} \mathbf{\Sigma}^{(l-1)} \mathbf{V}^\top \cdots \mathbf{U} \mathbf{\Sigma}^{(1)} \mathbf{V}^\top \mathbf{k}^{(0)}$$
$$= \mathbf{U} \mathbf{\Sigma}^{(l)} \mathbf{\Sigma}^{(l-1)} \cdots \mathbf{\Sigma}^{(1)} \mathbf{V}^\top \mathbf{k}^{(0)}$$

Then we know that the smallest singular value of $\mathbf{k}^{(l)}$ is determined by the product's smallest singular value. According to our assumption on the smallest singular value with respect to layer number, i.e. $\sigma_0^{(l)} = 1 - \alpha^l$, we obtain that:

$$\prod_{i=1}^{l} \sigma_0^{(l)} = \prod_{i=1}^{l} (1 - \beta^l).$$

We would like to calculate the limit value as the depth of GNN tends to be infinity:

$$\lim_{l \to \infty} \prod_{i=1}^{l} \sigma_0^{(l)} = \text{QPochhammer}[\beta]$$

which is a Quantum Pochhammer Symbol, and marked as QPochhammer. According to the property of Quantum Pochhammer Symbol, when $0 < \beta < 1$, we have that $\text{QPochhammer}[\beta] > 0$. As a result, we claim that the smallest eigenvalue of GNTK is greater than zero as depth tends to be infinity:

$$\lim_{l \to \infty} \lambda_0(\mathbf{K}^{(l)}) > 0$$

which completes the proof.

$\square$

## F  DATASET DESCRIPTION

We conclude the dataset statistics in Tab. 4 .

**Table 4:** Graph datasets statistics.

| Dataset | Task Type | Graphs | Nodes | Edges | Ave. Degree | Features | Classes | Metric |
|---------|-----------|--------|-------|-------|-------------|----------|---------|--------|
| Cora | Node Classification | 1 | 2,708 | 5,429 | 3.88 | 1,433 | 7 | Accuracy |
| Citeseer | Node Classification | 1 | 3,327 | 4,732 | 1.10 | 3,703 | 6 | Accuracy |
| PubMed | Node Classification | 1 | 19,717 | 44,338 | 8.00 | 500 | 3 | Accuracy |
| Ogbn-ArXiv | Node Classification | 1 | 169,343 | 1,166,243 | 13.77 | 128 | 40 | Accuracy |
| Ogbn-Proteins | Node Classification | 1 | 132,534 | 39,561,252 | 597.00 | 8 | 2 | ROC-AUC |
| Ogbn-Products | Node Classification | 1 | 2,449,029 | 61,859,140 | 50.52 | 100 | 47 | Accuracy |
| Ogbl-Collab | Link Prediction | 1 | 235,868 | 1,285,465 | 10.90 | 128 | 2 | Hits@50 |

## G  DETAILS ON EXPERIMENT CONFIGURATIONS

**Metrics.**  Accuracy represents the ratio of correctly predicted outcomes to the total predictions made. The ROC-AUC (Receiver Operating Characteristic-Area Under the Curve) value quantifies the probability that a randomly selected positive example will have a higher rank than a randomly selected negative example. Hit@50 denotes the proportion of correctly predicted edges among the top 50 candidate edges.

**Sparsity Ratio.**  We define the graph sparsity ratio $s_g$ and weight sparsity ratio $s_\theta$ in a single graph as follows:

$$s_g = \frac{1}{L} \sum_{l=0}^{L-1} \left( 1 - \frac{\left\| \boldsymbol{m}_{\mathrm{A}}^{(l)} \right\|_0}{\|\mathbf{A}\|_0} \right), \ s_\theta = \frac{1}{L} \sum_{l=0}^{L-1} \left( 1 - \frac{\left\| \boldsymbol{m}_{\theta}^{(l)} \right\|_0}{\|\boldsymbol{\Theta}^{(1)}\|_0} \right), \tag{14}$$

where $\| \cdot \|_0$ represents the $L_0$ norm counting the number of non-zero elements. In other words, $s_g$ represents the percentage of pruned edges out of all edges in the entire $L$ layers, while $s_\theta$ signifies the percentage of pruned elements out of all elements in the complete set of weights.

**Train-val-test Splitting of Datasets.**  For node classification in small- and medium-scale datasets, following the semi-supervised settings (Chen et al., 2021b), we utilized 140 labeled data points (Cora), 120 (Citeseer), and 60 (PubMed) for training, with 500 nodes allocated for validation and 1000 nodes for testing. In the context of deep GNNs, we take the full-supervised settings (Rong et al., 2019), which set 1000 nodes for testing, 500 nodes for validation and the others for training. The data splits for Ogbn-ArXiv, Ogbn-Proteins, Ogbn-Products, and Ogbl-Collab were provided by the benchmark (Hu et al., 2020). Specifically, for Ogbn-ArXiv, we train on papers published until 2017, validate on papers from 2018 and test on those published since 2019. For Ogbn-Proteins, protein nodes were segregated into training, validation, and test sets based on their species of origin. For Ogbn-Products, we sort the products according to their sales ranking and use the top 8% for training, next top 2% for validation, and the rest for testing. For Ogbl-Collab, we employed collaborations until 2017 as training edges, those in 2018 as validation edges, and those in 2019 as test edges.

**Backbone settings.**  For node classification and link prediction tasks on small- and medium-scale graphs, we employ 2-layer GCN/GIN/GAT backbones within the framework of standard-depth GNNs. In the context of deeper GNNs, we utilize 4/8/12/16-layer GCN/ResGCN/GAT backbones. For large-scale graphs including Ogbn-Arxiv/Proteins and Ogbl-Collab, we adopt DeeperGCN (Li et al., 2020a) of 4/12/20/28 layers. We choose 4/8-layer Cluster-GCN for Ogbn-Products.

For comparison with state-of-the-art GLT methods, we choose UGS (Chen et al., 2021b) and TGLT (Hui et al., 2023), which are the most efficient GLT methods to our best knowledge. For UGS, we directly adopt the source code provided by the authors and stick to their original parameter settings. For TGLT, we carefully reproduced their model from the description in the paper.

**Hyperparamters.**  We conclude the detailed hyperparameter settings in Tab. 5.

## H  ADDITIOANL EXPERIMENTS TO ANSWER RQ1

Fig. 9 showcases the results of `AdaGLT` on Cora dataset with the GCN/GIN/GAT backbone.

**Table 5:** Detailed hyper-parameter configurations. $\eta_g$ and $\eta_\theta$ denotes the coefficient attached to $\mathcal{R}(\boldsymbol{t}_A)$ and $\mathcal{R}(\boldsymbol{t}_\theta)$, respectively.

| Dataset | Model | Epochs (train/retain) | Optimizer | learning rate | Weight Decay | $\eta_g$ | $\eta_\theta$ | $\omega$ |
|---|---|---|---|---|---|---|---|---|
| | GCN | 400/300 | Adam | 0.01 | 8e-5 | 5e-5 | 0.001 | |
| Cora | GIN | 400/300 | Adam | 0.002 | 8e-5 | 2e-6 | 0.002 | 3 |
| | GAT | 400/300 | Adam | 0.001 | 8e-5 | 0 | 0.001 | |
| | GCN | 400/300 | Adam | 0.01 | 5e-4 | 9e-5 | 0.002 | |
| Citeseer | GIN | 400/300 | Adam | 0.002 | 5e-4 | 2e-6 | 0.002 | 2 |
| | GAT | 400/300 | Adam | 0.001 | 5e-4 | 0 | 0.001 | |
| | GCN | 500/400 | Adam | 0.01 | 5e-4 | 5e-5 | 0.001 | |
| PubMed | GIN | 300/300 | Adam | 0.002 | 5e-4 | 2e-6 | 0.002 | 3 |
| | GAT | 500/400 | Adam | 0.001 | 5e-4 | 0 | 0.001 | |
| Ogbn-Arxiv | | 800/600 | Adam | 0.01 | 0 | 1e-8 | 1e-12 | |
| Ogbn-Proteins | DeeperGCN | 200/150 | Adam | 0.01 | 0 | 7e-8 | 3e-13 | 3 |
| Ogbl-Collab | | 800/500 | Adam | 0.01 | 0 | 1e-8 | 1e-12 | |
| Ogbn-Product | Cluster-GCN | 50/50 | Adam | 0.001 | 0 | 1e-8 | 5e-12 | 2 |

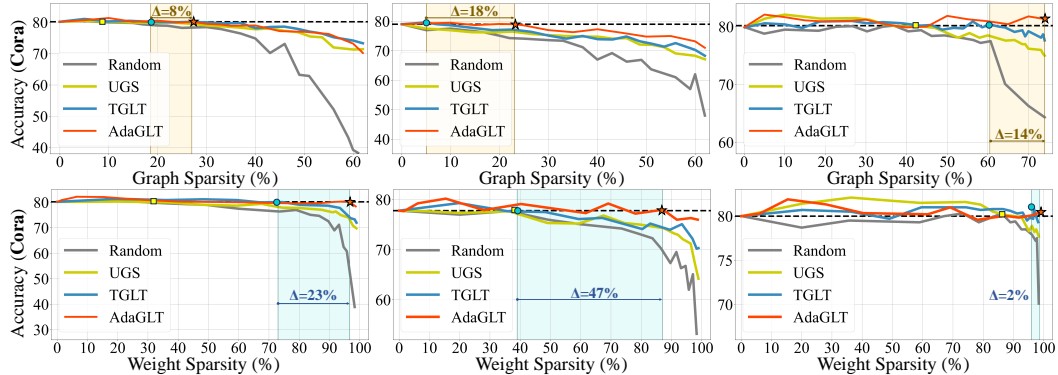

**Figure 9:** Results of node classification over Cora with GCN/GIN/GAT backbones. Black dash lines represent the baseline performance. Marker ■, ● and ★ indicates the last GLT that reaches higher accuracy than the original model in the sparsification process of UGS, TGLT and `AdaGLT`, respectively.

# I   ADDITIOANL EXPERIMENTS TO ANSWER RQ2

We provide detailed experimental results of the performance comparison between TGLT and `AdaGLT` on deep GAT/GCN/ResGCN (4 → 16 layers) in Tab. 6 to 10.

**Table 6:** The performance comparison between TGLT and `AdaGLT` in discovering GLTs on GAT backbone across various weight sparsity settings (10% → 90%) and GNN layer configurations (4 → 16 layers). Cells highlighted in red and blue correspond to winning tickets found by TGLT and `AdaGLT`, respectively.

| Weight Sparsity | Method | Cora | | | | Citeseer | | | | PubMed | | | |
|---|---|---|---|---|---|---|---|---|---|---|---|---|---|
| | | 4 | 8 | 12 | 16 | 4 | 8 | 12 | 16 | 4 | 8 | 12 | 16 |
| 0% | Baseline | 78.20 | 78.08 | 76.12 | 75.53 | 69.82 | 67.50 | 67.40 | 67.56 | 78.10 | 76.82 | 76.30 | 76.81 |
| 10% | TGLT | 78.79 | 79.39 | 73.52 | 71.49 | 70.11 | 67.79 | 68.10 | 67.08 | 78.64 | 77.91 | 77.09 | 76.92 |
| | AdaGLT | 79.12 | 78.27 | 76.84 | 75.98 | 69.90 | 68.67 | 68.11 | 67.99 | 78.43 | 78.19 | 77.76 | 77.09 |
| 30% | TGLT | 78.25 | 78.80 | 73.28 | 70.66 | 69.94 | 66.34 | 63.52 | 63.90 | 78.35 | 76.93 | 76.49 | 73.66 |
| | AdaGLT | 79.07 | 78.65 | 77.58 | 76.90 | 69.73 | 68.06 | 68.64 | 68.70 | 78.24 | 77.89 | 77.10 | 77.11 |
| 50% | TGLT | 78.21 | 76.42 | 70.19 | 72.00 | 69.86 | 67.73 | 64.49 | 60.70 | 78.13 | 74.36 | 72.97 | 70.28 |
| | AdaGLT | 78.14 | 78.29 | 76.89 | 75.00 | 69.93 | 67.74 | 67.44 | 67.63 | 78.14 | 76.98 | 76.08 | 76.23 |
| 70% | TGLT | 73.74 | 73.65 | 70.56 | 70.88 | 67.26 | 63.40 | 60.77 | 62.89 | 74.48 | 69.08 | 68.71 | 65.58 |
| | AdaGLT | 78.89 | 78.14 | 72.70 | 70.85 | 69.88 | 67.07 | 67.63 | 66.92 | 79.04 | 77.41 | 73.19 | 72.43 |
| 90% | TGLT | 70.10 | 67.71 | 62.28 | 63.56 | 64.22 | 63.19 | 55.46 | 54.76 | 66.80 | 67.72 | 59.71 | 60.02 |
| | AdaGLT | 75.83 | 76.41 | 69.90 | 63.18 | 64.83 | 64.50 | 62.68 | 60.13 | 76.03 | 72.28 | 69.58 | 67.46 |

**Table 7:** The performance comparison between TGLT and `AdaGLT` in discovering GLTs on GCN backbone across various graph sparsity settings (10% → 60%) and GNN layer configurations (4 → 16 layers).

| Graph Sparsity | Method | Cora | | | | Citeseer | | | | PubMed | | | |
|---|---|---|---|---|---|---|---|---|---|---|---|---|---|
| | | 4 | 8 | 12 | 16 | 4 | 8 | 12 | 16 | 4 | 8 | 12 | 16 |
| 0% | Baseline | 83.95 | 83.65 | 84.80 | 83.60 | 75.10 | 74.20 | 73.80 | 74.00 | 88.10 | 84.40 | 85.60 | 83.10 |
| 10% | TGLT | 78.09 | 78.74 | 76.08 | 75.09 | 69.90 | 67.83 | 67.62 | 66.88 | 78.32 | 76.92 | 85.79 | 78.22 |
| | AdaGLT | 84.25 | 84.87 | 84.82 | 81.28 | 75.58 | 74.80 | 76.21 | 74.19 | 89.55 | 85.60 | 85.92 | 84.01 |
| 20% | TGLT | 84.02 | 79.05 | 78.9 | 77.15 | 67.06 | 68.04 | 66.30 | 66.93 | 83.30 | 85.07 | 81.44 | 71.27 |
| | AdaGLT | 85.55 | 79.15 | 78.68 | 76.45 | 75.23 | 72.20 | 69.40 | 68.23 | 88.14 | 84.77 | 85.92 | 77.68 |
| 30% | TGLT | 79.55 | 71.06 | 70.50 | 66.55 | 66.15 | 63.21 | 59.60 | 58.62 | 85.87 | 82.22 | 79.70 | 73.18 |
| | AdaGLT | 81.76 | 77.75 | 75.30 | 69.15 | 75.45 | 70.70 | 70.21 | 60.40 | 89.14 | 85.59 | 85.73 | 75.01 |
| 40% | TGLT | 73.15 | 70.04 | 70.88 | 67.37 | 64.55 | 61.50 | 51.38 | 43.40 | 78.41 | 72.98 | 73.44 | 70.62 |
| | AdaGLT | 82.25 | 75.82 | 70.00 | 66.13 | 74.15 | 71.60 | 69.00 | 54.31 | 88.33 | 84.57 | 80.61 | 76.18 |
| 50% | TGLT | 67.75 | 68.15 | 62.92 | 58.65 | 58.84 | 54.17 | 30.72 | 37.40 | 72.16 | 70.18 | 68.24 | 66.79 |
| | AdaGLT | 77.65 | 71.97 | 67.82 | 60.27 | 73.46 | 70.20 | 70.71 | 52.23 | 81.17 | 83.64 | 75.15 | 73.29 |
| 60% | TGLT | 60.65 | 45.89 | 38.69 | 43.25 | 51.07 | 23.81 | 16.50 | 14.30 | 66.18 | 65.16 | 64.04 | 54.88 |
| | AdaGLT | 75.17 | 71.45 | 64.49 | 53.25 | 73.55 | 70.40 | 65.44 | 57.87 | 78.92 | 75.43 | 70.48 | 67.44 |

**Table 8:** The performance comparison between TGLT and `AdaGLT` in discovering GLTs on GCN backbone across various weight sparsity settings (10% → 90%) and GNN layer configurations (4 → 16 layers).

| Weight Sparsity | Method | Cora | | | | Citeseer | | | | PubMed | | | |
|---|---|---|---|---|---|---|---|---|---|---|---|---|---|
| | | 4 | 8 | 12 | 16 | 4 | 8 | 12 | 16 | 4 | 8 | 12 | 16 |
| 0% | Baseline | 83.95 | 83.65 | 84.80 | 83.60 | 75.10 | 74.20 | 73.80 | 74.00 | 88.10 | 84.40 | 85.60 | 83.10 |
| 10% | TGLT | 80.35 | 80.85 | 82.43 | 81.85 | 71.45 | 71.59 | 72.80 | 75.00 | 88.73 | 86.42 | 83.77 | 80.32 |
| | AdaGLT | 85.55 | 85.27 | 85.30 | 82.75 | 76.55 | 75.20 | 76.31 | 74.39 | 88.23 | 85.07 | 85.72 | 84.79 |
| 30% | TGLT | 81.35 | 81.67 | 83.69 | 83.68 | 70.46 | 72.00 | 72.11 | 74.23 | 88.29 | 82.09 | 85.78 | 82.94 |
| | AdaGLT | 85.34 | 85.65 | 84.68 | 81.35 | 76.35 | 74.91 | 75.82 | 74.88 | 89.21 | 85.22 | 85.88 | 83.71 |
| 50% | TGLT | 78.94 | 78.97 | 80.03 | 81.65 | 68.95 | 72.20 | 71.22 | 73.39 | 86.07 | 84.57 | 82.22 | 79.50 |
| | AdaGLT | 85.75 | 84.53 | 83.90 | 81.15 | 76.65 | 75.00 | 76.32 | 74.00 | 88.63 | 85.67 | 86.48 | 83.11 |
| 70% | TGLT | 77.45 | 75.85 | 79.08 | 76.84 | 65.72 | 68.17 | 67.80 | 66.91 | 78.21 | 78.92 | 76.40 | 76.23 |
| | AdaGLT | 84.83 | 84.15 | 80.80 | 78.25 | 75.53 | 74.20 | 75.68 | 73.10 | 88.92 | 86.04 | 86.39 | 84.15 |
| 90% | TGLT | 75.35 | 70.95 | 70.51 | 67.26 | 64.55 | 62.08 | 53.20 | 58.61 | 76.40 | 76.11 | 72.27 | 70.09 |
| | AdaGLT | 85.43 | 83.55 | 81.70 | 78.55 | 76.23 | 74.50 | 70.58 | 64.50 | 89.38 | 84.67 | 85.34 | 82.86 |

**Table 9:** The performance comparison between TGLT and `AdaGLT` in discovering GLTs on ResGCN backbone across various graph sparsity settings (10% → 60%) and GNN layer configurations (4 → 16 layers).

| Graph Sparsity | Method | Cora | | | | Citeseer | | | | PubMed | | | |
|---|---|---|---|---|---|---|---|---|---|---|---|---|---|
| | | 4 | 8 | 12 | 16 | 4 | 8 | 12 | 16 | 4 | 8 | 12 | 16 |
| 0% | Baseline | 84.40 | 84.15 | 84.25 | 82.00 | 75.20 | 74.40 | 74.15 | 73.45 | 87.30 | 85.50 | 84.40 | 87.20 |
| 10% | TGLT | 80.20 | 79.15 | 79.54 | 80.25 | 73.21 | 72.25 | 73.29 | 69.88 | 87.49 | 85.91 | 84.69 | 86.74 |
| | AdaGLT | 85.42 | 84.47 | 81.45 | 82.75 | 75.40 | 76.27 | 75.11 | 75.35 | 88.21 | 86.68 | 84.96 | 87.81 |
| 20% | TGLT | 79.70 | 77.85 | 76.75 | 76.95 | 69.50 | 67.75 | 71.17 | 63.23 | 87.05 | 82.66 | 80.47 | 82.65 |
| | AdaGLT | 85.32 | 82.15 | 79.55 | 81.94 | 74.60 | 74.25 | 72.70 | 72.23 | 87.18 | 85.81 | 84.66 | 86.98 |
| 30% | TGLT | 73.40 | 72.25 | 72.23 | 69.28 | 64.51 | 61.75 | 64.60 | 61.05 | 85.82 | 77.26 | 80.06 | 79.82 |
| | AdaGLT | 82.39 | 82.15 | 79.25 | 82.65 | 74.31 | 74.85 | 72.11 | 73.15 | 87.67 | 85.37 | 84.86 | 87.14 |
| 40% | TGLT | 72.60 | 71.45 | 71.35 | 68.15 | 68.92 | 70.95 | 61.72 | 57.15 | 80.62 | 78.39 | 78.50 | 77.49 |
| | AdaGLT | 80.83 | 78.35 | 78.45 | 77.88 | 74.13 | 71.65 | 73.10 | 74.20 | 87.48 | 85.92 | 85.01 | 86.88 |
| 50% | TGLT | 72.90 | 73.35 | 74.95 | 64.77 | 60.31 | 66.55 | 61.32 | 63.55 | 75.87 | 75.14 | 72.32 | 69.56 |
| | AdaGLT | 79.11 | 76.55 | 75.25 | 75.75 | 70.40 | 74.47 | 72.26 | 71.85 | 83.90 | 85.86 | 84.68 | 86.25 |
| 60% | TGLT | 62.90 | 70.15 | 65.45 | 69.26 | 59.80 | 66.35 | 66.35 | 63.26 | 68.03 | 70.64 | 66.29 | 68.05 |
| | AdaGLT | 75.10 | 73.55 | 72.75 | 74.15 | 71.90 | 73.06 | 71.13 | 71.37 | 83.81 | 82.19 | 82.12 | 83.61 |

**Table 10:** The performance comparison between TGLT and `AdaGLT` in discovering GLTs on ResGCN backbone across various weight sparsity settings (10% → 90%) and GNN layer configurations (4 → 16 layers). Cells highlighted in red and blue correspond to winning tickets found by TGLT and `AdaGLT`, respectively.

| Weight Sparsity | Method | Cora | | | | Citeseer | | | | PubMed | | | |
|---|---|---|---|---|---|---|---|---|---|---|---|---|---|
| | | 4 | 8 | 12 | 16 | 4 | 8 | 12 | 16 | 4 | 8 | 12 | 16 |
| 0% | Baseline | 84.40 | 84.15 | 84.25 | 82.00 | 75.20 | 74.40 | 74.15 | 73.45 | 87.30 | 85.50 | 84.40 | 87.20 |
| 10% | TGLT | 79.79 | 81.45 | 82.54 | 80.16 | 71.00 | 72.87 | 75.70 | 72.88 | 86.89 | 85.55 | 84.52 | 86.95 |
| | AdaGLT | 87.71 | 86.47 | 84.64 | 82.61 | 75.58 | 76.25 | 74.31 | 74.25 | 88.04 | 86.45 | 85.67 | 87.44 |
| 30% | TGLT | 79.40 | 80.85 | 83.05 | 80.57 | 69.76 | 72.35 | 73.40 | 71.03 | 86.08 | 85.69 | 84.72 | 86.14 |
| | AdaGLT | 86.00 | 86.16 | 83.55 | 81.22 | 74.43 | 75.20 | 74.14 | 74.03 | 87.56 | 85.89 | 85.10 | 88.04 |
| 50% | TGLT | 79.23 | 79.12 | 79.83 | 80.25 | 69.21 | 70.45 | 73.30 | 70.44 | 83.18 | 80.58 | 82.30 | 84.30 |
| | AdaGLT | 85.98 | 85.08 | 84.15 | 82.03 | 75.31 | 73.35 | 73.50 | 73.04 | 88.23 | 85.69 | 84.98 | 87.68 |
| 70% | TGLT | 77.60 | 77.85 | 76.75 | 76.99 | 70.52 | 68.80 | 71.79 | 65.21 | 83.12 | 80.81 | 79.58 | 80.20 |
| | AdaGLT | 85.20 | 84.14 | 83.77 | 80.82 | 73.53 | 73.53 | 73.00 | 72.25 | 87.12 | 85.74 | 84.76 | 87.03 |
| 90% | TGLT | 72.60 | 71.14 | 72.28 | 69.97 | 68.92 | 58.76 | 64.38 | 59.33 | 80.33 | 78.12 | 78.60 | 74.75 |
| | AdaGLT | 84.73 | 83.85 | 82.66 | 79.98 | 73.08 | 73.85 | 74.44 | 73.71 | 87.78 | 85.34 | 84.66 | 86.94 |

## J  ADDITIOANL EXPERIMENTS TO ANSWER RQ3

Fig. 10 to 13 comprehensively illustrate the performance comparison of `AdaGLT` with TGLT, UGS, and random pruning on DeeperGCN at 4, 12, 20, and 28 layers, across Ogbn-Arxiv, Ogbn-Proteins, and Ogbl-Collab datasets. Fig. 14 illustrates the sparsity distribution of GLTs uncovered by `AdaGLT` under different settings and datasets.

Tab. 11 demonstrate the performance of `AdaGLT` on Ogbn-Products with 4- and 8-layer Cluster-GCN. It can be observed that (1) `AdaGLT` can scale up to large graphs like Ogbn-Products and effectively find GLTs, while UGS completely fails. (2) Discovering winning tickets within shallow GNNs is more feasible. Under 30% graph sparsity, `AdaGLT` successfully identifies winning tickets at a 4-layer GCN, while struggling to achieve baseline performance on an 8-layer GCN. This observation aligns with our findings in Sec. 4.3.

**Table 11:** The performance of AdaGLT on Ogbn-Products with 4- and 8-layer Cluster-GCN. We report the mean accuracy ± stdev of 3 runs.

| Layer | 4 (Baseline=$79.23_{\pm0.47}$) | | | 8 (Baseline=$78.82_{\pm0.73}$) | | |
|---|---|---|---|---|---|---|
| **Graph Sparsity** | 10% | 30% | 50% | 10% | 30% | 50% |
| UGS | $78.44_{\pm0.58}$ | $74.67_{\pm0.62}$ | $73.05_{\pm1.02}$ | $76.30_{\pm0.79}$ | $72.13_{\pm1.25}$ | $71.32_{\pm1.14}$ |
| AdaGLT | $80.35_{\pm0.51}$ | $79.67_{\pm0.86}$ | $76.46_{\pm0.69}$ | $80.22_{\pm0.78}$ | $78.13_{\pm1.12}$ | $74.62_{\pm0.90}$ |

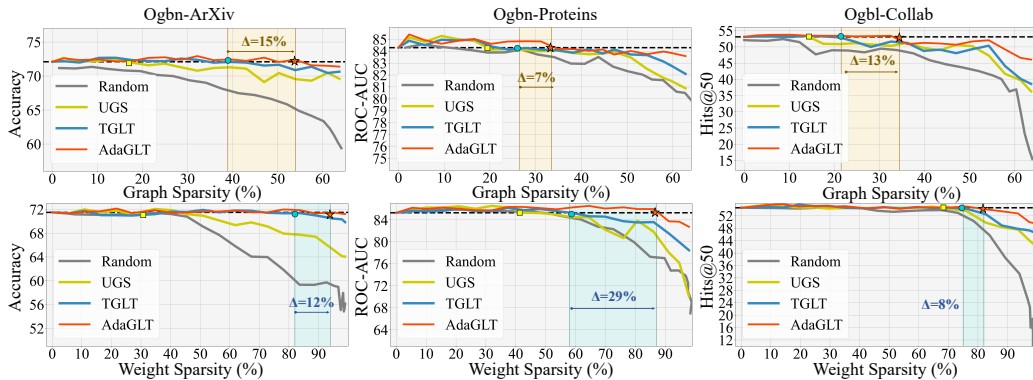

**Figure 10:** The performance comparison on Ogbn-Arxiv/Ogbn-Proteins/Ogbl-Collab datasets with 4-layer DeeperGCN. Black dash lines represent the baseline performance.

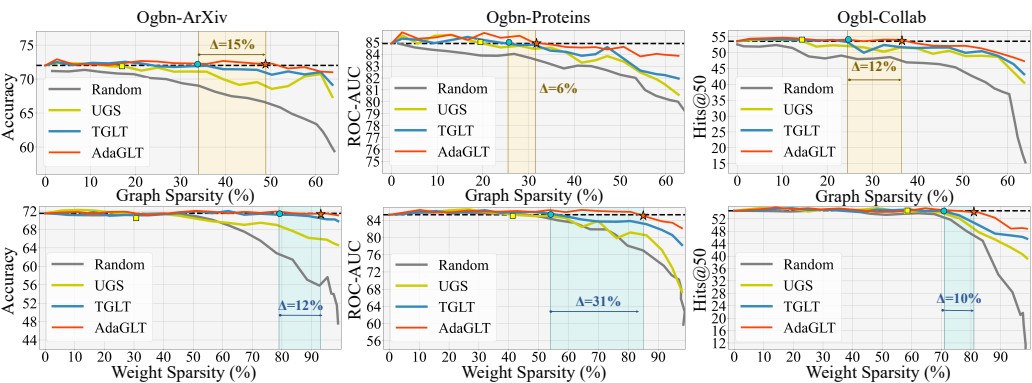

**Figure 11:** The performance comparison on Ogbn-Arxiv/Ogbn-Proteins/Ogbl-Collab datasets with 12-layer DeeperGCN. Black dash lines represent the baseline performance.

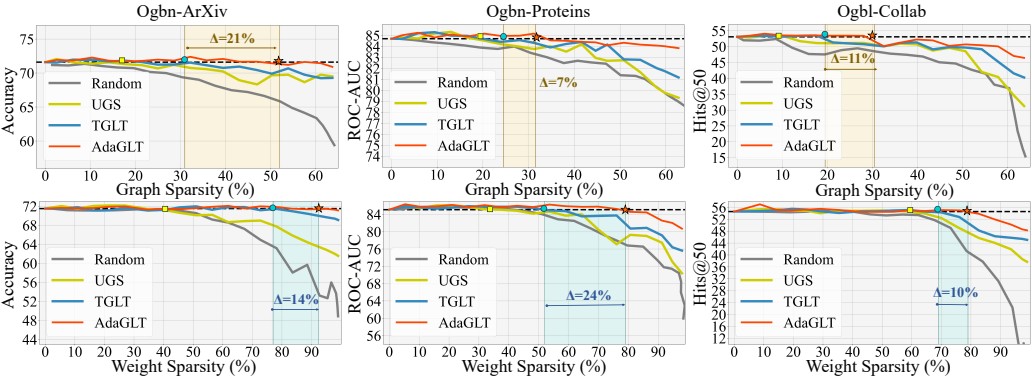

**Figure 12:** The performance comparison on Ogbn-Arxiv/Ogbn-Proteins/Ogbl-Collab datasets with 20-layer DeeperGCN. Black dash lines represent the baseline performance.

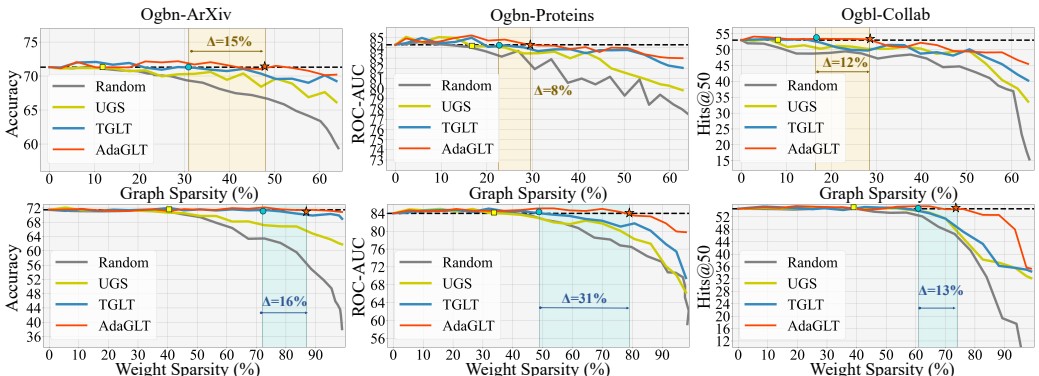

**Figure 13:** The performance comparison on Ogbn-Arxiv/Ogbn-Proteins/Ogbl-Collab datasets with 28-layer DeeperGCN. Black dash lines represent the baseline performance.

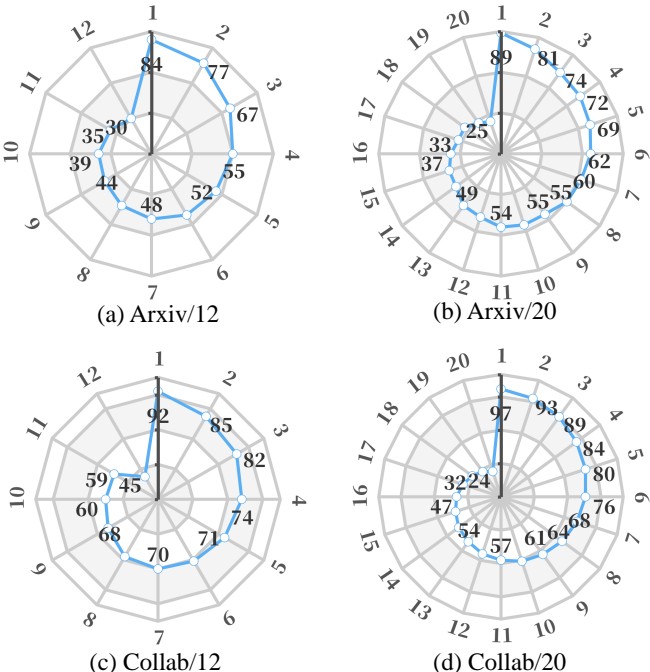

**Figure 14:** (a) denotes the percentage of remaining edges at each layer of a 12-layer DeeperGCN on Ogbn-Arxiv after applying `AdaGLT`. (b) denotes that under 20-layer settings with Ogbn-Arxiv. (c)/(d) illustrates that under 12/20-layer settings with Ogbl-Collab.

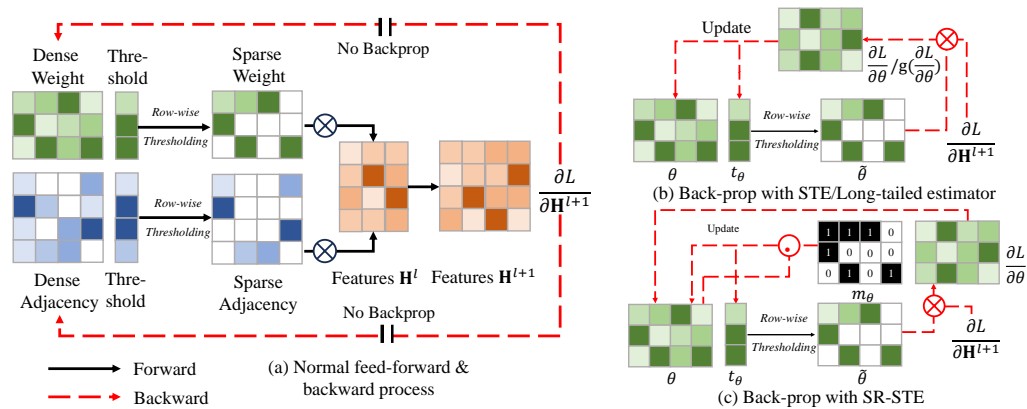

**Figure 15:** In this figure, $\odot$ denotes element-wise multiplication and $\otimes$ indicates matrix multiplication. (a) demonstrates the forward and backward processes of simultaneous sparsification without gradient estimators. The backpropagation process is blocked due to the binarization operation. (b) depicts the weight and its associated threshold vector updates using straight-through/long-tailed estimator, and $g(\cdot)$ denotes the long-tail transformation (Xu & Cheung, 2019). (c) elucidates the weight and its associated threshold vector updates using SR-STE.

## K  ABLATION STUDY ON GRADIENT ESTIMATORS

We have employed the straight-through estimator (STE) to enable differentiable binary pruning mask calculations. However, there exist alternative gradient estimators. To assess the sensitivity of `AdaGLT` to different gradient estimators, we have also incorporated the LTE (Long-tailed estimator) (Xu & Cheung, 2019; Liu et al., 2020) and SR-STE (Zhou et al., 2021). Fig. 15 (a) illustrates the blocked backpropagation process when joint sparsification is introduced into the graph convolution, while (b) details how STE and LTE update weights and threshold vectors. In the forward stage, $\tilde{\Theta}$ is obtained by the row-wise thresholding with threshold vector $\mathbf{t}_\theta$. And in the backward stage, the gradient w.r.t. $\tilde{\Theta}$ will be applied to $\Theta$ directly. Fig. 15 (c) demonstrates how SR-STE updates weights and threshold vectors. Different from STE, $\Theta$ and $t_\theta$ in this case are updated not only through $\frac{\partial \mathcal{L}}{\partial \tilde{\Theta}}$ but also guided by $\boldsymbol{m}_\theta \odot \boldsymbol{\Theta}$, leading to a more stable sparsification process (Zhou et al., 2021).

We conducted a comprehensive comparison of various gradient estimators applied to `AdaGLT`. Tab. 12 presents the extreme graph sparsity achieved by each gradient estimator, which corresponds to the most sparse graph lottery ticket found. Tab. 13 displays the extreme weight sparsity obtained by each gradient estimator. It is evident that STE and LTE demonstrate consistent performance across various datasets and GNN models. However, it is noteworthy that SR-STE tends to exhibit suboptimal performance, particularly on GCN and GIN. On GIN, SR-STE achieved a maximum graph sparsity decrease of merely 6.8% and a weight sparsity decrease of 7.3%. This can be attributed to the fact that SR-STE was introduced to prevent the dynamic sparsifying procedure from ineffectively alternating the pruned network architecture (Zhou et al., 2021). Paradoxically, this hinders `AdaGLT` from exploring a wider range of sparse structures.

**Table 12:** The extreme graph sparsity at which `AdaGLT` is able to find GLTs with different gradient estimators on different datasets and backbones.

| Estimators | Dataset + Model | | | | | |
|---|---|---|---|---|---|---|
| | Cora +GCN | Cora +GIN | Cora +GAT | Citeseer +GCN | Citeseer +GIN | Citeseer +GAT |
| STE | 27.6 | 22.7 | 74.1 | 45.7 | 42.0 | 84.2 |
| LTE | 28.9 | 21.2 | 71.8 | 47.2 | 44.0 | 85.9 |
| SR-STE | 22.7 | 15.9 | 72.8 | 43.8 | 37.9 | 83.3 |

**Table 13:** The extreme weight sparsity at which `AdaGLT` is able to find GLTs with different gradient estimators on different datasets and backbones.

| Estimators | Dataset + Model | | | | | |
|---|---|---|---|---|---|---|
| | Cora +GCN | Cora +GIN | Cora +GAT | Citeseer +GCN | Citeseer +GIN | Citeseer +GAT |
| STE | 96.7 | 87.0 | 98.7 | 96.7 | 96.2 | 98.9 |
| LTE | 97.4 | 87.2 | 98.3 | 97.2 | 95.4 | 98.7 |
| SR-STE | 95.8 | 79.9 | 96.8 | 93.4 | 93.9 | 97.1 |

**Table 14:** Ablation study on Citeseer with GCN backbone of $2 \rightarrow 8$ layers and $20\% \sim 60\%$ graph sparsity, evaluating the edge explainer (EE) and layer-adaptive pruning (LP). "w/o EE" signifies the replacement of the edge explainer with a trainable mask, and "w/o LP" indicates the maintenance of the same sparsity across all layers. The underlined number denotes the highest performance under certain graph sparsity across all `AdaGLT` variants.

| Layer | 2 | | | 4 | | | 6 | | | 8 | | |
|---|---|---|---|---|---|---|---|---|---|---|---|---|
| Sparsity | 20% | 40% | 60% | 20% | 40% | 60% | 20% | 40% | 60% | 20% | 40% | 60% |
| `AdaGLT` | 71.32 | 70.66 | 66.23 | 75.23 | 74.36 | 72.93 | 74.49 | 71.46 | 71.12 | 71.33 | 69.27 | 69.42 |
| `AdaGLT` w/o EE | 68.69 | 54.16 | 48.05 | 52.60 | 45.77 | 42.28 | 53.17 | 47.10 | 40.85 | 52.91 | 40.66 | 40.66 |
| `AdaGLT` w/o LP | 71.47 | 71.15 | 65.90 | 75.50 | 71.46 | 65.99 | 69.74 | 67.10 | 62.83 | 67.05 | 60.79 | 55.63 |

**Table 15:** Ablation study on Citeseer with GAT backbone of $2 \rightarrow 8$ layers and $40\% \sim 80\%$ graph sparsity, evaluating the edge explainer (EE) and layer-adaptive pruning (LP).

| Layer | 2 | | | 4 | | | 6 | | | 8 | | |
|---|---|---|---|---|---|---|---|---|---|---|---|---|
| Sparsity | 40% | 60% | 80% | 40% | 60% | 80% | 40% | 60% | 80% | 40% | 60% | 80% |
| `AdaGLT` | 70.12 | 70.05 | 70.09 | 69.83 | 68.13 | 66.47 | 68.46 | 66.12 | 66.43 | 67.50 | 64.98 | 63.29 |
| `AdaGLT` w/o EE | 67.44 | 61.46 | 57.15 | 62.48 | 57.90 | 53.48 | 54.91 | 45.12 | 45.12 | 43.10 | 43.10 | 43.10 |
| `AdaGLT` w/o LP | 70.13 | 70.23 | 70.06 | 68.72 | 66.45 | 66.30 | 65.94 | 64.17 | 64.08 | 62.33 | 60.74 | 56.58 |

## L ADDITIONAL ABLATION EXPERIMENTS

To validate the effectiveness of the individual components, we conduct additional ablation experiments in this section, focusing on two pivotal components of `AdaGLT`: the edge explainer and layer-adaptive pruning. Specifically, we aim to address the following two questions:

1. What is the impact on performance when our proposed edge explainer is substituted with a trainable mask employed in Chen et al. (2021b)?

2. How does performance vary when maintaining the same sparsity level for each layer compared to increasing sparsity as the number of layers grows?

We chose Citeseer + GCN/GAT (for small graphs) and Ogbn-Arxiv + DeeperGCN (for large graphs) for ablation study. Our experimental results are shown in Tab. 14 to 16. We list two straightforward observations:

**Obs.1.** Substituting the edge explainer resulted in a significant performance decline. Specifically, in deeper GNNs, such as 6- and 8-layer GCN/GAT, replacing the edge explainer with a trainable mask rendered the network unable to identify any winning tickets.

**Obs.2.** Layer-adaptive pruning significantly aids in discovering winning tickets in deeper GNNs. We observe that in 2- and 4-layer GCNs, AdaGLT without layer-adaptive pruning occasionally outperforms original AdaGLT. However, in deep scenarios, the elimination of layer-adaptive pruning results in a sharp performance decline (8.48%↓ under 40% sparsity and 13.79%↓ under 60% sparsity in an 8-layer GCN).

**Table 16:** Ablation study on Ogbn-Arxiv with DeeperGCN backbone of $4 \rightarrow 28$ layers and 60% graph sparsity, evaluating the edge explainer (EE) and layer-adaptive pruning (LP).

| Layer | 4 | 12 | 20 | 28 |
|---|---|---|---|---|
| AdaGLT | 70.13 | 70.08 | 71.34 | 69.79 |
| AdaGLT w/o EE | 60.34 | 52.60 | 47.44 | 42.40 |
| AdaGLT w/o LP | 68.74 | 63.89 | 60.33 | 58.05 |

