# OpenReview forum: "Graph Lottery Ticket Automated"
_ICLR.cc/2024/Conference — ICLR 2024 poster_

### Official Review · Reviewer_9nRP · 2023-10-29

**Soundness:** 3 good
**Presentation:** 3 good
**Contribution:** 2 fair
**Rating:** 6
**Confidence:** 4

**Summary:**

This paper argues that existing graph lottery ticket algorithms like UGS heavily rely on trial-and-error pruning rate tuning and scheduling, and suffer from stability issues when extended to deeper GCNs. Those limitations call for an adaptive framework to automatically identify the pruning hyper-parameters that improve the scalability of GCNs. To this end, the authors propose a framework called AdaGLT, adaptive graph lottery tickets, to overcome such limitations.
Extensive experiments validate the effectiveness of the proposed AdaGLT as compared to previous ad-hoc UGS.

**Strengths:**

1. The paper is well-written and organized. As compared to previous UGS, the proposed AdaGLT jointly sparsifies both graphs and GNN weights and adopts an adaptive layer sparsification process. In addition, it offers the practitioner the chance to adopt automatic pruning and dynamic restoration for extracting the graph lottery tickets.

2. The effective combination of automation, layer adaptivity, and dynamic pruning/restoration provides better properties as compared to previous methods.

**Weaknesses:**

Given the contribution of UGS and other series of graph lottery tickets work, this automated tool sounds like incremental work. However, the analysis of scalability issues when applied to deep GNNs is worth reading.

Most experiments are conducted on small and transductive graph/tasks. The ogbn-arxiv is a small graph and ogbn-protein is a medium graph actually but are claimed as large graphs, more reasonablely large graphs should be considered. Also, how about inductive settings like GraphSAGE or other SOTA ones?

Other than the above aspects, I think the other perspectives of this paper are clear.

**Questions:**

See weaknesses. I wonder whether the contribution applied to an inductive setting or evolving graphs with increasing nodes or edges.

---

> ### Author Response · Authors · 2023-11-15
> **[Part 1] Response to Reviewer 9nRP**
>
> We express our sincere thanks for the detailed and thoughtful review of our manuscript and for the encouraging appraisal of our work. We give point-by-point responses to your comments and describe the revisions we made to address them:
>
> ## **Weakness 1: Given the contribution of UGS and other series of graph lottery tickets work, this automated tool sounds like incremental work. However, the analysis of scalability issues when applied to deep GNNs is worth reading.**
>
> Thank you for recognizing our analysis of GLT and deep GNNs! However, we would like to clarify the significance of finding an "automated" ticket: In fact, the impact of the pruning ratios, $p_g$ (graph pruning ratio) and $p_\theta$ (weight pruning ratio), on GLT performance is more substantial than conventionally understood in previous GLT methods.
>
> **Table 1.** The performance of 10 rounds iterative UGS on Citeseer + GCN with different $p_g$  and $p_\theta$.
> | $p_g$ \ $p_θ$ | 0.05  | 0.10  | 0.15  | 0.20  | 0.25  |
> |-------|-------|-------|-------|-------|-------|
> | **0.05**  | 69.08 | 69.63 | 70.34 | 69.83 | 68.27 |
> | **0.10**  | 64.23 | 67.88 | 67.97 | 66.99 | 66.14 |
> | **0.15**  | 60.23 | 62.55 | 65.43 | 63.45 | 60.96 |
> | **0.20**  | 59.61 | 61.28 | 62.70 | 61.38 | 60.44 |
> | **0.25**  | 59.08 | 60.92 | 61.44 | 62.17 | 57.60 |
>
> **Table 2.** The performance of 10 rounds iterative UGS on PubMed + GIN with different $p_g$ and $p_\theta$.
> | $p_g$ \ $p_θ$ | 0.05  | 0.10  | 0.15  | 0.20  | 0.25  |
> |-------|-------|-------|-------|-------|-------|
> | **0.05**  | 77.30|78.40 |  76.79|  76.92| 76.25|
> | **0.10**  |  76.09 | 76.18 | 76.44 | 76.04 |75.18 |  |
> | **0.15**  | 75.33| 74.93|  75.80|75.66  | 73.12|  |
> | **0.20**  | 75.57|76.60 | 75.30 | 74.78 | 73.58 |  |
> | **0.25**  | 72.18| 73.18| 73.62 | 70.44 |68.40 |  |
>
> We observe that (1) traditional GLT methods are significantly influenced by the values of $p_g$ and $p_\theta$, resulting in substantial performance disparities, reaching as high as $12.74$%. (2) the original setting of UGS ($p_g = 0.05, p_\theta = 0.20$) is not universally optimal;in the case of Citeseer + GCN, the optimal combination is {$p_g = 0.05, p_\theta = 0.15$}, and for Pubmed + GIN, it is {$p_g = 0.05, p_\theta = 0.10$}.
>
> In this context, the pursuit of an "automated" GLT that is free of pruning ratio tuning becomes particularly meaningful.
>
> ## **Weakness 2: …more reasonablely large graphs should be considered. Also, how about inductive settings like GraphSAGE or other SOTA ones?**
>
> Thanks for your thoughtful comments that make our results evens stronger! Verifying the performance of AdaGLT on larger graphs or in an inductive setting is indeed crucial. We will respond to your inquiries in three aspects:
>
> - How is the performance of AdaGLT on Ogbn-Products (2,449,029 nodes and 61,859,140 edges) ?
> - How can we (slightly) modify AdaGLT to facilitate its migration to inductive tasks?
> - How does AdaGLT perform on other inductive tasks like graph classification?
>
>
> **1. Performance with Ogbn-Products**
>
> **Experimental Setup.** We opt for Ogbn-Products with Cluster-GCN to validate AdaGLT on a larger graph. Dataset splitting is consistent with that in [3]. The reason for not selecting DeeperGCN, as in the other three OGB datasets, is the excessively time-consuming training of DeeperGCN on Ogbn-Products (nearly 20h/run with NVIDIA V100 32GB).
>
>
> **Table 3.** The performance of AdaGLT on Ogbn-Products with 4- and 8- layer Cluster-GCN. We report the average of 3 runs.
> | Layer | 4 (Baseline=79.23±0.47) | | | 8 (Baseline=78.82±0.73) | | |
> | --- | --- | --- | --- | --- | --- | --- |
> | Graph Sparsity | 10% | 30% | 50% | 10% | 30% | 50% |
> | UGS | 78.44±0.58 | 74.67±0.62 | 73.05±1.02 | 76.30±0.79 | 72.13±1.25 | 71.32±1.14 |
> | AdaGLT | 80.35±0.71 | 79.67±0.86 | 76.46±0.69 | 80.22±0.78 | 78.13±1.12 | 74.62±0.90 |
>
>
> We list two straightforward observations:
>
> **Obs. 1.** AdaGLT can scale up to large graphs like Ogbn-Products and effectively find GLTs, while UGS completely fails.
>
> **Obs. 2.** Discovering winning tickets within shallow GNNs is more feasible. Note that under $30\%$ graph sparsity, AdaGLT successfully identifies winning tickets at a 4-layer GCN, while struggling to achieve baseline performance on an 8-layer GCN. This observation aligns with our findings in Section 4.3 Obs. 4 and Appendix J.

---

> ### Author Response · Authors · 2023-11-15
> **[Part 2] Response to Reviewer 9nRP**
>
> (continued response to Weakness 2)
>
> **2. Modification for inductive tasks**
>
> We will first (1) discuss why conventional GLT methods [1,2] cannot be extended to inductive settings; (2) explain how the components of AdaGLT can be slightly modified for inductive settings.
>
> Allow us to respectfully recall that, UGS introduces two differentiable masks $m_A$ and $m_\theta$, with shapes identical to $\mathbf{A}$ and $\mathbf{\Theta}$. This decides that UGS is inherently transductive and cannot be applied to scenarios with multiple graphs and varying numbers of nodes. In contrast, AdaGLT adopts an edge explainer to generate a soft edge mask of size $\mathcal{O}(E)$, which means that it can generate edge masks for multiple graphs of arbitrary sizes.
>
> However, another component of AdaGLT, namely the trainable thresholds, hinders its direct transition to inductive settings. It is noteworthy that, for graph pruning, we employ a threshold vector $t^{(l)}_A \in \mathbb{R}^{N}$ for each layer $l$. However, such fixed-length thresholds are inherently transductive.
>
> This obstacle is readily addressed: akin to the transition from a trainable mask $m_A$ in UGS to an edge explainer $\Psi$ in AdaGLT, we can seamlessly replace trainable thresholds with generated ones. Specifically, we employ a simple MLP $\Upsilon$ in each layer to generate the trainable threshold:
>
> $t_A^{(l)} = \operatorname{Sigmoid}\left(\Upsilon^{(l)} (\mathbf{X})\right) \in \mathbb{R}^{N},\;l\in\{0,1,\cdots,L-1\}$.
>
> Note that $\mathbf{X}$ is the node feature matrix and $\Upsilon^{(l)}$ followed by a sigmoid function transform $\mathbf{X}$ into the node-wise thresholds at layer $l$. Practically, we leverage a 2-layer MLP with hidden dimensions {$\{64,1\}$} in the next section.
>
> This modification to AdaGLT involves simply replacing Line 6 in the Algorithm table with the following one:
>
>
> $t_A^{(l)} = \operatorname{Sigmoid}\left(\Upsilon^{(l)} (X)\right) , m_{{A},ij}^{(l)} = \mathbb{1} [{s_{ij} < t_{A,i}^{(l)}}] \prod_{k=0}^{l-1}m_{{A},ij}^{(k)}$
>
> With both the edge masks and thresholds **generated** rather than being fixed parameters, AdaGLT can be seamlessly incorporated in any inductive scenario. We employ this version of AdaGLT in the subsequent section to report its performance.
>
>
>
> **3. Performance in graph classification**
>
> **Experimental Setup.** We choose MNIST[3] and MUTAG datasets to validate AdaGLT's scalability to graph classification tasks. Following [4], we split MNIST dataset to 55000 train/5000 validation/10000 test. Following [5], we perform 10-fold cross-validation on MUTAG.
>
> **Table 4.** The performance of AdaGLT on MNIST with 3-layer GraphSAGE (Baseline=**93.40**) compared with random pruning. We report an average of 5 runs.
> | Graph Sparsity  | 20% | 30% | 40%  | 50% | 60%  |
> |-------|-------|-------|-------|------|-------|
> | Random |89.16±0.76 | 87.81±0.58 | 79.46±0.89 | 71.48±1.12 | 67.20±3.64 |
> | AdaGLT  | 94.88±0.83 | 93.53±1.12 | 93.47±1.33 | 92.22±1.79 | 89.64±1.66 |
>
> **Table 5.** The performance of AdaGLT on MUTAG with 3-layer GraphSAGE (Baseline=**85.78**) compared with random pruning. We report the average of 10-fold cross-validation.
> | Graph Sparsity  | 20% | 30% | 40%  | 50% | 60%  |
> |-------|-------|-------|-------|------|-------|
> | Random  | 84.22±3.49 | 83.78±4.08 | 79.26±3.59 | 73.11±7.65| 65.28±9.40 |
>  AdaGLT| 86.76±2.71 | 86.30±2.96 | 85.41±2.44 | 83.23±3.79  | 79.55±4.27|
>
> It is noteworthy that AdaGLT extends its impressive performance from node classification to graph classification. Specifically, AdaGLT successfully identifies winning tickets with 40% and 30% graph sparsity on MNIST and MUTAG, respectively.
>
> ## **Question 1: I wonder whether the contribution applied to an inductive setting or evolving graphs with increasing nodes or edges.**
>
> We validate the performance of AdaGLT in graph classification in response to Weakness 2 :)
>
> **Summary**:
>
> Your valuable insights have significantly contributed to advancing our work! In the newly uploaded draft, we have made the following revisions and the revised or newly incorporated text is highlighted in blue:
>
> - Our response to Weakness 2.1, concerning the results of AdaGLT with Ogbn-Product, is now provided in Appendix J.
> - Our response to Weakness 2.2, regarding modifications for inductive settings, is introduced in Appendix M.
> - Our response to Weakness 2.3, focusing on AdaGLT for graph classification, is incorporated into Appendices G and H.
>
> **References**:
>
> [1] A Unified Lottery Ticket Hypothesis for Graph Neural Networks, ICML'21
>
> [2] Rethinking Graph Lottery Tickets: Graph Sparsity Matters, ICLR'23
>
> [3] Open graph benchmark: Datasets for machine learning on graphs, NeurIPS'20
>
> [4] Benchmarking graph neural networks, JLMR'22
>
> [5] How powerful are graph neural networks?, ICLR'19

---

> ### Author Response · Authors · 2023-11-22
> **Respectful Inquiry Before Discussion Deadline**
>
> Dear Reviewer 9nRP,
>
> Hope this message finds you well. As the author-reviewer discussion deadline approaches, we respectfully seek your confirmation on the adequacy of our rebuttal in addressing the concerns raised in your review.
>
> We are rather gratified to receive your insightful response! Your feedback on the **scalability to larger graphs** and **inductive setting** has significantly enhanced the quality of our manuscript, and we have diligently addressed and responded to your concerns in both the OpenReview response and the updated manuscript.
>
> We really appreciate the substantial time commitment on your part and extend heartfelt thanks for any additional insights. Your expertise is instrumental in refining our projects and positively impacting our research endeavors.
>
> Thank you once again for your time and valuable perspectives. We eagerly await your further guidance with utmost respect.
>
> Best regards,
>
> Authors

---

### Official Review · Reviewer_9csi · 2023-10-31

**Soundness:** 3 good
**Presentation:** 3 good
**Contribution:** 3 good
**Rating:** 6
**Confidence:** 3

**Summary:**

The authors proposed the AdaGLT to obtain the graph lottery ticket to sparsify both the trainable weights and the graph structure together. The AdaGLT learns different masks for different layers of graph and weights, providing higher freedom of the sparsification. To save the memory for the training mask, the edge explainer was also introduced. Through extensive evaluation, the proposed framework achieves satisfactory results.

**Strengths:**

1. The evaluation is sufficient to support the claims.
2. The paper presentation is clear enough to get the main ideas.
3. AdaGLT can work on deep GNNs and large-scale datasets.

**Weaknesses:**

1. Section 3.1 is the direct application of Liu et al. 2022, as stated in the paper. The edge explainer is also available. So, the main contribution seems incremental.
2. The assumption of Theorem 1 may not be true for $G^{(l)}$ in most cases, since $G^{(l)}$ should be different due to the layer-adaptive pruning.
3. The algorithm does not include the "Dynamic Pruning and Restoration" paragraph. And the statement of equation 10 is unclear to me. What does the restoration refer to?

**Questions:**

1. What does the "irreversible fashion" refer to? The baseline UGS updates the mask only, and both the original A and weights are stored, so it should be considered as "reversible"?
2. The authors stated that "Existing GLT methods .... and their lack of flexibility stems from the necessity of hyperparameter tuning". Can the author explain more in detail? The baseline UGS also has the trainable mask, which can be considered as the other version of the trainable threshold.
3. Could the author describe how to get the equation 10?

---

> ### Author Response · Authors · 2023-11-15
> **[Part 1] Response to Reviewer 9csi**
>
> Thank you for your thoughtful and constructive reviews of our manuscript! Based on your questions and recommendations, we give point-by-point responses to your comments and describe the revisions we made to address them. For better reading and comprehension, we slightly rearrange the order of your comments.
>
> ## **Weakness 2: The assumption of Theorem 1 may not be true for $G^{(l)}$ in most cases, since $G^{(l)}$ should be different due to the layer-adaptive pruning.**
>
> Thanks for your insightful question. We would like to clarify that $\mathbf{G}^{(l)}$ differs across the layer index $l$. In our setting, the key distinction is in the smallest singular value $\sigma^{(l)}_0 = 1 - \alpha^l$, which varies with the layer index. This setting of the smallest singular value is in accordance with *layer-adaptive pruning*. Additionally, other singular values can also differ within $\mathbf{G}^{(l)}$. Consequently, the operation matrix $\mathbf{G}^{(l)}$ varies from layer to layer.
>
> ## **Question 1: What does the "irreversible fashion" refer to? The baseline UGS updates the mask only, and both the original A and weights are stored, so it should be considered as "reversible"?**
>
> Really sorry for bringing confusion! "Irreversible" refers to the fact that, in the iterative magnitude pruning process of UGS, the elements (either edge or weight) pruned at the end of each iteration (i.e., setting the mask $m_{ij}$ to zero) will not have an opportunity to be restored in the next iteration (i.e., $m_{ij}$ will never be set back to one).
>
> For a more detailed elucidation, we would like to respectfully outline the pruning logic of UGS (take **graph pruning** for example):
>
> - Iteration 1: Train GNN with $\mathcal{G}=\{m_g \odot \mathbf{A},\mathbf{X}\}$ for $n$ epochs. Set $p_g\%$ of the lowest values in $m_g$ to zero, thus obtaining $m_g^{(1)}$. We denote $m_g \setminus m_g^{(1)}$ as the pruned edges in iteration 1.
> - Iteration 2: Train GNN with $\mathcal{G}=\{m_g^{(1)} \odot \mathbf{A},\mathbf{X}\}$ for another $n$ epochs. Use the same pruning ratio $p_g\%$ and obtain $m_g^{(2)}$. $m_g^{(1)} \setminus m_g^{(2)}$ denotes the pruned edges in iteration 2.
> - This iterative process continues as shown in our Figure 2 ($\textit{Right}$).
>
> It is crucial to note that $m_g^{(k-1)} \setminus m_g^{(k)}$ that is pruned in Iteration $k$ will not be reactivated in any subsequent iteration: **once pruned, they are irreversibly set to zero, which represents what we term an "irreversible fashion."**
>
> Adhering to the conventional LTH, UGS resets the GNN weights to their initialization values at the start of each iteration, which corresponds with the recoverability of initialization (in a **reversible** way).  However, when we refer to "**irreversible**", we specifically mean the irrecoverability of the pruned elements.
>
>
> ## **Weakness 3: The algorithm does not include the "Dynamic Pruning and Restoration" paragraph. And the statement of equation 10 is unclear to me. What does the restoration refer to?**
>
> Thanks for pointing out this issue! We would like to clarify: the characteristic of *dynamic pruning* in AdaGLT is not a specifically designed module; rather, it is a natural result of our decision to discard **fixed pruning ratios** in favor of using **trainable thresholds**. In the algorithm table (Appendix C), the dynamic pruning is done in Lines 5-6, and restoration is done in Lines 9-10. Namely, the gradient updates of $\mathbf{\Theta}, \Psi, t_\theta$, and $t_A$ implicitly entail the recovery of certain edges or weights.
>
> With this in mind, we would like to offer a more explicit explanation of Equation 10 and how the **restoration** is done:
>
> Take weight pruning for example. In epoch $k$, $W \in \mathbb{R}^{N \times M}$ is a learnable matrix. Given certain $i,j$, if $W_{ij}^{(k)} < t_i^{(k)}$, then $m_{ij}^{(k)} = 0$, which means $W_{ij}^{(k)}$ is **pruned** in epoch $k$. However, possibilities exist that $W_{ij}$ can be **recovered** or **restored**, namely $m_{ij}^{(k+1)} = 1$, in epoch $k+1$, as long as the updated $W_{ij}^{(k+1)}$ is greater than $t_i^{(k+1)}$. In other words:
>
> $W_{ij}^{(k+1)} = W_{ij}^{(k)} - \alpha\cdot\text{grad w.r.t } W_{ij}^{(k)} $ > $t_{i}^{(k)} - \alpha\cdot\text{grad w.r.t } t_{i}^{(k)} =t_i^{(k+1)} $
>
> In the original Equation 10, we rearrange the expression as follows:
>
> $W_{ij}^{(k)} - t_i > \alpha\cdot(\text{grad w.r.t } W_{ij}^{(k)}- \text{grad w.r.t } t_{i}^{(k)})$
>
> The content following the equality sign in Equation 10 directly exhibits the computed gradients, derived in part from Equation 4.
>
> Hopefully, our explanation of the restoration process provides a clearer understanding of how we achieve "dynamic restoration". We have modified Appendix C to better illustrate the **adaptive, dynamic, and automated** nature of our algorithm.

---

> ### Author Response · Authors · 2023-11-15
> **[Part 2] Response to Reviewer 9csi**
>
> ## **Question 2: The authors stated that "Existing GLT methods .... and their lack of flexibility stems from the necessity of hyperparameter tuning". Can the author explain more in detail? The baseline UGS also has the trainable mask, which can be considered as the other version of the trainable threshold.**
>
> Yes, we are glad to clarify! Allow us to give a straightforward explanation of why conventional GLT methods are characterized by a ``lack of flexibility stemming from the necessity of hyperparameter tuning``. As shown in Table 1, the performance of UGS is hugely influenced by the combination of $p_g$ and $p_\theta$, and the original setting ($p_g = 0.05, p_\theta=0.20$) is actually not the optimal hyperparameter configurations for Citeseer + GCN. On the contrary, our proposed AdaGLT completely eliminates the need for adjusting $p_g$ and $p_\theta$, and it is capable of automatically adjusting how many elements to prune or recover during different training stages.
>
> **Table 1.** The performance of 10 rounds iterative UGS on Citeseer + GCN with different $p_g$ (pruning ratio for graph) and $p_\theta$ (pruning ratio for weight).
> | $p_g$ \ $p_θ$ | 0.05  | 0.10  | 0.15  | 0.20  | 0.25  |
> |-------|-------|-------|-------|-------|-------|
> | **0.05**  | 69.08 | 69.63 | 70.34 | 69.83 | 68.27 |
> | **0.10**  | 64.23 | 67.88 | 67.97 | 66.99 | 66.14 |
> | **0.15**  | 60.23 | 62.55 | 65.43 | 63.45 | 60.96 |
> | **0.20**  | 59.61 | 61.28 | 62.70 | 61.38 | 60.44 |
> | **0.25**  | 59.08 | 60.92 | 61.44 | 62.17 | 57.60 |
>
> Regarding your concern that ``The baseline UGS also has the trainable mask, which can be considered as the other version of the trainable threshold``, we would like to respectfully clarify that, though the $m_A$ and $m_\theta$ in UGS are indeed trainable, they are pruned in an "untrainable" manner: it is up to practitioners to manually define the pruning ratios. And we believe this brought inconvenience in real-world applications.
>
>
> ## **Question 3: Could the author describe how to get the equation 10?**
>
> Please refer to our response to Weakness 3 :)
>
>
> ## **Weakness 1: Section 3.1 is the direct application of Liu et al. 2022, as stated in the paper. The edge explainer is also available.**
>
> Thanks for your appreciation of our presentations and experimental designs!
>
> Please allow us to systematically elaborate on the connections and differences between our work and previous ones:
>
> **1. Trainable thresholds**
>
> **Table 2.** The comparison of trainable thresholds utilized in AdaGLT and previous work.
> |  |  **AdaGLT**   | **A-ViT[6]**  |
> |-------|-------|-------|
> |**Threshold Level** | Threshold Scalar/**Vector**/Matrix | Threshold **Scalar** |
> | **Gradient Estimator** |  STE/SR-STE/LTE |STE |
> | **Pruning Object**| Edge & Weight | Attention token |
> | **High Sparsity** | Yes | No |
>
> As shown in Table 2, the thresholds employed in AdaGLT exhibit notable distinctions from those in [5] in at least the following four aspects:
>
> -  **Threshold Level**: We thoroughly discuss and compare the performance of thresholds at different levels, ultimately selecting the threshold vector, which is not presented in [5]. (in Section 4.5 and Appendix B)
>
> - **Gradient Estimator**: We extensively discuss and compare various gradient estimators, confirming that AdaGLT is not sensitive to the choice of estimator and ultimately selecting STE, a comprehensive analysis absent in [5]. (in Section 4.5 and Appendix K)
>
> - **Pruning Object**: The pruning targets in AdaGLT differ significantly from those in [5]. The former involves joint pruning of $A$ and $\mathbf{\Theta}$, while the latter focuses on pruning attention tokens. It is noteworthy that applying trainable thresholds to edges with intrinsic values of $\{0, 1\}$ is non-trivial. To address this challenge, we introduced the edge explainer to provide dense edge scores. (in Section 3.2)
>
> - **High Sparsity**: The thresholds in [5] cannot achieve the high sparsity desired for GLT. Consequently, we introduced sparse regularization tailored to our application scenario. (in Section 3.3)

---

> ### Author Response · Authors · 2023-11-15
> **[Part 3] Response to Reviewer 9csi**
>
> (continued response to **Weakness 1**)
>
> **2. Edge Explainer**
>
>
> **Table 3.** The comparison of edge explainer utilized in AdaGLT and previous GNN explanation methods.
> |  |  **AdaGLT**   | **PGexplainer[6]**  | **GNNExplainer[7]** |
> |-------|-------|-------|------|
> |**Stage** | Within training | Post-hoc | Post-hoc
> |**Parameterized** | Yes (Eq.6) | Yes (MLP) | No |
> | **Differentiable Trick** |  Gradient estimator (STE/SR-STE/LTE) | Reparameterization trick | N/A
> | **Target**| Prune edges | Sample explantory graphs | Sample explantory graphs |
> | **Layer-adaptive**|Yes |No | No|
> | **Towards Higher Pruning Rtio**|Sparse Regularization |Budget constraint | No|
>
>
> As shown in Table 3, our edge explainer is different from previous edge explainers in the following aspects:
>
> - **Stage**: Our edge explainer is concurrently trained with network parameters. In contrast to post-hoc explanations in [6,7], our approach allows for better integration with the network, yielding more robust explanation scores.
> - **Parameterized**: Our edge explainer is paramterized as [6], but is with different implementations.
> - **Differentiable Trick**: Sampling edges from the edge scores provided by edge explainers is inherently not differentiable, and the methods we adopted are different from those in [6,7].
> - **Target & Layer-adaptive**: The utilization of edge explainer in prior studies typically aimed at extracting explanatory subgraphs (note that here, "subgraph" refers to *the same structure across all layers*). In contrast, we are the first to explore the use of edge explainer in the domain of graph sparsification to achieve *layer-adaptive* pruning.
> - **High Pruning Ratio**: [7] does not explicitly ensure a high sparsity rate for the subgraph, and [6] imposes a budget constraint to obtain a compact explanatory graph. However, they fall short of achieving high graph sparsity. In contrast, AdaGLT achieves state-of-the-art sparsity with no performance compromise through sparse regularization.
>
>
>
> **Summary**:
>
> We have diligently addressed each of the concerns raised, and your comments have significantly improved the manuscript. Hopefully, our responses and revisions meet your expectations for a higher rating.
>
>
> **References:**
>
> [1] A Unified Lottery Ticket Hypothesis for Graph Neural Networks, ICML'21
>
> [2] Adversarial Erasing with Pruned Elements: Towards Better Graph Lottery Tickets, ECAI'23
>
> [3] Rethinking Graph Lottery Tickets: Graph Sparsity Matters, ICLR'23
>
> [4] Pruning graph neural networks by evaluating edge properties, KBS'22
>
> [5] Adaptive Sparse ViT: Towards Learnable Adaptive Token Pruning by Fully Exploiting Self-Attention, IJCAI'23
>
> [6] Parameterized explainer for graph neural network, NeurIPS'20
>
> [7] Gnnexplainer: Generating explanations for graph neural networks, NeurIPS'19
>
> [7] Interpretable and Generalizable Graph Learning via Stochastic Attention Mechanism, ICML'22

---

> > ### Comment · Reviewer_9csi · 2023-11-18
> >
> > The authors addressed my major concerns, so I increased my score.

---

> > > ### Author Response · Authors · 2023-11-20
> > > **Response to Reviewer 9csi**
> > >
> > > We thank the reviewer 9csi for more strongly supporting our paper! We are glad our revision and rebuttal have addressed your concerns！

---

### Official Review · Reviewer_2ZB4 · 2023-11-05

**Soundness:** 3 good
**Presentation:** 3 good
**Contribution:** 3 good
**Rating:** 5
**Confidence:** 3

**Summary:**

The paper proposes Adaptive, Dynamic, and Automated framework for identifying Graph Lottery Tickets to overcome the integration of
GLT into deeper and larger-scale GNN contexts. It attempts tailoring layer-adaptive sparse structures for various datasets and GNNs, thus endowing it with the capability to facilitate deeper GNNs; integrating the pruning and training processes, thereby achieving a dynamic workflow encompassing both pruning and restoration; and automatically capturing graph lottery tickets across diverse sparsity levels, obviating the necessity for extensive pruning parameter tuning.

**Strengths:**

1. The paper is well written with motivation explicitly explained. The evolution of both edges and weights might dynamically during the training process as well as the flexibility of prune ratio are important missing links that are explored.
2. The introduction of an edge explainer into GNN pruning that ensures interpretability during the pruning process while reducing unimportant edges, is a good way to mitigate the quadratic increase in parameters.
3. Appendix is rich. I will recommend the authors move some large-scale experiments on OGBN-Arxiv and Products to be moved to main draft.

**Weaknesses:**

Although the introduced components like join sparsification, edge explainer etc make sense, one major concern I have is how these individual components affect the performance of AdaGLT. I appreciate the authors for their extensive experiments, but the role/importance of individual modules is not well explained. How does removing some components affect the performance. Another question, no significant performance benefit is observed for low sparsity ratios eg 30-50%. Any explanation of why it is effective only in high sparsity ratios will improve the manuscripts. I am open to increasing my score after the rebuttal discussion.

**Questions:**

See above.

---

> ### Author Response · Authors · 2023-11-11
> **[Part 1] Response to Reviewer 2ZB4**
>
> We sincerely thank you for your careful comments and thorough understanding of our paper! Here we give point-by-point responses to your comments and describe the revisions we made to address them. We would be happy to add additional clarifications and revisions to the paper to address any additional recommendations.
>
> > **Comment 1: I will recommend the authors move some large-scale experiments on OGBN-Arxiv and Products to be moved to main draft.**
>
> Thanks for your suggestions! In our newly submitted manuscript, we replace the experimental results in Figure 5 with the performances of AdaGLT applied to Arxiv/Collab/Proteins on 12-layer DeeperGCN for graph sparsity.
>
> > **Comment 2: Although the introduced components like join sparsification, edge explainer etc make sense, one major concern I have is how these individual components affect the performance of AdaGLT. I appreciate the authors for their extensive experiments, but the role/importance of individual modules is not well explained. How does removing some components affect the performance.**
>
> Thank you for pointing out this issue. In the current draft, we indeed lack ablation experiments regarding the importance of some components. From our perspective, the following components of AdaGLT are amenable to ablation (**Ab.**) experiments:
>
> **Ab.1.** Edge explainer: What is the impact on performance if the edge explainer is replaced by a trainable mask employed in UGS?
>
> **Ab.2.** Layer-adaptive pruning: How does performance change if the same sparsity level is maintained for each layer, as opposed to increasing sparsity with the growth of layers?
>
> **Ab.3.** [Solved] Gradient estimator: What is the effect of different gradient estimators on AdaGLT's performance? (Refer to Section 4.5 and Appendix K)
>
> **Ab.4.** [Solved] Threshold level: How do trainable thresholds at different levels affect AdaGLT's performance? (Refer to Section 4.5 and Appendix B)
>
> Automated pruning (without tuning pruning ratios) is another crucial component in our work. However, replacing it with fixed pruning rates is not a trivial task (as it requires setting different pruning rates for each layer and discarding the gradient estimator). Therefore, we do not consider it here.
>
> To address your concerns regarding component ablation, we select Pubmed (for small graphs) and Ogbn-Arxiv (for large graphs) to complement the experiments for Ab1 and Ab2.
>
>
> **Table 1.** Ablation study on Citeseer with GCN backbone of $2 \rightarrow 8$ layers and $20 \rightarrow 60$ graph sparsity, evaluating the edge explainer (EE) and layer-adaptive pruning (LP). 'w/o EE' signifies the replacement of the edge explainer with a trainable mask, and 'w/o LP' indicates the maintenance of the same sparsity across all layers. The **bold** number denotes the highest performance under certain graph sparsity across all AdaGLT variants.
> | Layer | 2         |          |          | 4         |          |          | 6         |          |          | 8         |          |          |
> |-------|-----------|----------|----------|-----------|----------|----------|-----------|----------|----------|-----------|----------|----------|
> |   Graph Sparsity  | 20%       | 40%      | 60%      | 20%       | 40%      | 60%      | 20%       | 40%      | 60%      | 20%       | 40%      | 60%     |
> | AdaGLT   | 71.32      | 70.66    | **66.23**    | 75.23     |**74.36**    | 72.93    | **74.49**     | **71.46**    | **71.12**    | **71.33**     | **69.27**    | **69.42**   |
> | AdaGLT w/o EE | 68.69      | 54.16    | 48.05    | 52.60     | 45.77    | 42.28    | 53.17     | 47.10    | 40.85    | 52.91     | 40.66    | 40.66   |
> | AdaGLT w/o LP | **71.47**     | **71.15**    | 65.90    | **75.50**    |71.46    | 65.99    | 69.74     | 67.10*   | 62.83    | 67.05     | 60.79    | 55.63    |
>
>
> **Table 2** Ablation study on Ogbn-Arxiv with DeeperGCN backbone of $4 \rightarrow 28$ layers and $60$% graph sparsity, evaluating the edge explainer (EE) and layer-adaptive pruning (LP).
> | Layer | 4    | 12   | 20   | 28   |
> |-------|------|------|------|------|
> | AdaGLT        | **70.13** |**70.08** |**71.34** | **69.79**|
> | AdaGLT w/o EE | 60.34 | 52.60 | 47.44 | 42.40 |
> | AdaGLT w/o LP | 68.74 | 63.89 | 60.33 | 58.05 |
>
> We list two straightforward observations:
>
> **Obs.1.**  Substituting the edge explainer resulted in a significant performance decline. Specifically, in deeper GNNs, such as 6- and 8-layer GCN/GAT, replacing the edge explainer with a trainable mask rendered the network unable to identify any winning tickets.
>
> **Obs.2.** Layer-adaptive pruning significantly aids in discovering winning tickets in deeper GNNs. We observe that in 2- and 4-layer GCNs, AdaGLT without layer-adaptive pruning occasionally outperforms original AdaGLT. However, in deep scenarios, the elimination of layer-adaptive pruning results in a sharp performance decline (8.48% under 40% sparsity and 13.79% under 60% sparsity in an 8-layer GCN).

---

> ### Author Response · Authors · 2023-11-11
> **[Part 2] Response to Reviewer 2ZB4**
>
> > **Comment 3: Another question, no significant performance benefit is observed for low sparsity ratios eg 30-50%. Any explanation of why it is effective only in high sparsity ratios will improve the manuscripts.**
>
> Thanks again for making our results even stronger! We encountered a similar problem when presenting our experimental results, and we are delighted to share additional insights with you (which are not included in the manuscript). Allow us to humbly recap the features of AdaGLT: To achieve a desired sparsity level, we apply Sparse Regularization to increase the values of the threshold vector (corresponding to higher sparsity, see Section 3.3). Consequently, even when aiming for GLTs of 30%~50% sparsity, the network tends to converge to 70% or even higher sparsity in the end (especially for certain backbones like GAT). This implies that the obtained $m_A$ at lower sparsity levels often does not receive sufficiently thorough training compared to higher sparsity levels, resulting in less significant improvements in found GLTs.
>
> To address this issue, we initially propose a straightforward solution: adjusting the coefficient $η_g$ of Sparse Regularization to control the network's convergence to different sparsity levels. Experimental results demonstrate that this approach can improve GLTs' performance at lower sparsity levels (as shown in the tables below).
>
> | Citeseer + GCN (average of 3 runs); baseline=70.08 | |||||
> |----------------------------------------------------|-----------|-----------|-----------|-----------|-----------|
> | sparsity                                                  | 20%       | 30%       | 40%       | 50%       | 60%       |
> | $\eta_g$                                              | 1e-3      | 1.2e-3    | 1.8e-3    | 1.8e-3    | 2e-3      |
> | Performance (with carefully tuned $n_g$)           | 72.04 ± 0.85 | 71.33 ± 1.12 | 71.48 ± 0.93 | 69.94 ± 1.34 | 68.03 ± 1.52 |
> | Performance (reported in the paper)                | 70.32     | 70.18     | 70.34     | 69.02     | 68.12     |
>
> | PubMed + GAT (average of 3 runs); baseline=78.5 | |||||
> |----------------------------------------------------|-----------|-----------|-----------|-----------|-----------|
> | sparsity                                                  | 20%       | 30%       | 40%       | 50%       | 60%       |
> | $\eta_g$                                            | 0.6e-3  | 0.75e-3 | 0.85e-3 | 1e-3  | 1e-3  |
> | Performance (with carefully tuned $\eta_g$)         | 80.24 ± 1.11 | 80.33 ± 1.52 | 80.03 ± 0.89 | 79.64 ± 1.87 | 79.88 ± 1.01 |
> | Performance (reported in the paper)              | 79.93 | 78.90 | 78.46 | 80.10 | 78.77 |
>
> However, we ultimately chose not to adopt this strategy, primarily for the following three reasons:
>
> - We aim to diverge from the conventional GLT methods that extensively adjust the pruning ratio, hence asserting our contribution to "automated pruning." Tuning the sparsity coefficient for optimal performance would introduce another form of hyperparameter tuning, which we seek to avoid.
> - While GLTs at lower sparsity levels did not exhibit significant improvements in certain setups, they showed notable enhancements in more settings (like GIN and GAT backbones in Figure 4).
> - Considering real-world applications, we contend that GLTs with higher sparsity levels hold more value.
>
> Based on these considerations, we present the manuscript as you have seen it, and hopefully this response meets your satisfaction.
>
> We have incorporated responses to Comments 1 & 2 in the updated manuscript. Thank you again for your thoughtful comments that greatly strengthen our paper! We would be delighted to address any further inquiries.

---

> ### Author Response · Authors · 2023-11-22
> **Respectful Inquiry Before Discussion Deadline**
>
> Dear Reviewer 2ZB4,
>
> We hope this message finds you well. As the deadline for the author-reviewer discussion phase is nearing, we would like to respectfully inquire if our rebuttal has effectively addressed the concerns raised in your review.
>
> Your insightful feedback, especially regarding the **ablation study** and **observations at low sparsity**, has been invaluable in refining our work. We have endeavored to address each of these points meticulously in our rebuttal and are eager to know if our responses and subsequent revisions have met your expectations and clarified the issues you pointed out.
>
> We deeply recognize the demands of your time and deeply appreciate any further feedback on our rebuttal. Your expertise is not only vital for the review process but also enriches our ongoing learning and growth in this field.
>
> Thank you once again for your time and invaluable insights. We eagerly await your response.
>
> Best regards,
>
> Authors

---

### Author Response · Authors · 2023-11-20
**Global Response to All Reviewers**

Dear reviewers,

Thank you again for your thoughtful and constructive comments! We are really encouraged to see that the reviewers appreciate some positive aspects of our paper, such as **technical novelty** (Reviewers 2ZB4, 9nRP), **theoretical guarantees** (Reviewer 9nRP), **thorough experimental validation** (Reviewers 2ZB4, 9csi, 9nRP) and **superior scalability** (Reviewer 9csi, 9nRP).

Your expertise significantly helps us strengthen our manuscript – this might be the most helpful review we have received in years! In addition to addressing your thoughtful comments point-to-point on OpenReview, we have made the following modifications to the newly uploaded manuscript (all updated text is highlighted in blue):

1. **To Reviewer 2ZB4**:
    - Following your Comment 1, we replaced the experimental results in Figure 5 with the performances of AdaGLT applied to Arxiv/Collab/Proteins on 12-layer DeeperGCN for graph sparsity;
    - Following your Comment 2, we have comprehensively supplemented the ablation study regarding edge explainer and layer-adaptive pruning in Appendix L.

2. **To Reviewer 9csi**:
    - Following your Weakness 3, we revised the Algorithm table for better comprehension in Appendix C.

3. **To Reviewer 9nRP**, following your Weakness 2:
    - we supplemented the results of AdaGLT on Ogbn-Product in Appendix J;
    - we elaborated on the minor modification of AdaGLT for inductive settings in Appendix M;
    - we supplemented the performance of AdaGLT for graph classification in Appendices G and H.

We have made earnest efforts to address the primary concerns raised, and sincerely appreciate the acknowledgment and improved rating from reviewer 9csi.  We also respectfully look forward to the thoughtful feedback from the reviewers to further enhance the quality of our manuscript.

---

### Author Response · Authors · 2023-11-23
**Gentle Reminder**

Dear Reviewers and Area Chair,

We hope this message finds you well. We would like to respectfully remind you that the author-reviewer discussion deadline is approaching in under five hours. We are always eager to hear your valuable feedback and provide further responses.

Warm regards,

Authors

---

### Meta-Review · Area_Chair_xjtR · 2023-12-18

**Metareview:**

This paper proposes a new strategy for pruning graph neural networks. This was a borderline paper; the reviewers liked the approach and found the results to be productive, but they were concerned about the incremental nature of the paper and the opacity of how much each of the proposed techniques contributed to the final method. I'm ambivalent about whether to accept the paper, and I would be open to acceptance or rejection.

**Justification For Why Not Higher Score:**

The paper is borderline. It's incremental and there are some questions about the rigor of the ablations.

**Justification For Why Not Lower Score:**

The technique does still represent a step forward and the reviewers agreed on that.

---

### Decision · Program_Chairs · 2024-01-16

Accept (poster)